# Prc1-rich kinetochores are required for error-free acentrosomal spindle bipolarization during meiosis I in mouse oocytes

Shuhei Yoshida[1], Sui Nishiyama[1,2], Lisa Lister[3,4], Shu Hashimoto[1,5,6], Tappei Mishina[1], Aurélien Courtois[1], Hirohisa Kyogoku[1], Takaya Abe [7], Aki Shiraishi[7], Meenakshi Choudhary[3,4], Yoshiharu Nakaoka[6], Mary Herbert[3,4] & Tomoya S. Kitajima [1,2✉]

Acentrosomal meiosis in oocytes represents a gametogenic challenge, requiring spindle bipolarization without predefined bipolar cues. While much is known about the structures that promote acentrosomal microtubule nucleation, less is known about the structures that mediate spindle bipolarization in mammalian oocytes. Here, we show that in mouse oocytes, kinetochores are required for spindle bipolarization in meiosis I. This process is promoted by oocyte-specific, microtubule-independent enrichment of the antiparallel microtubule cross-linker Prc1 at kinetochores via the Ndc80 complex. In contrast, in meiosis II, cytoplasm that contains upregulated factors including Prc1 supports kinetochore-independent pathways for spindle bipolarization. The kinetochore-dependent mode of spindle bipolarization is required for meiosis I to prevent chromosome segregation errors. Human oocytes, where spindle bipolarization is reportedly error prone, exhibit no detectable kinetochore enrichment of Prc1. This study reveals an oocyte-specific function of kinetochores in acentrosomal spindle bipolarization in mice, and provides insights into the error-prone nature of human oocytes.

---

[1] Laboratory for Chromosome Segregation, RIKEN Center for Biosystems Dynamics Research (BDR), Kobe 650-0047, Japan. [2] Graduate School of Biostudies, Kyoto University, 606-8501 Kyoto, Japan. [3] Biosciences Institute, Newcastle University, Centre for Life, Times Square, Newcastle upon Tyne NE1 4EP, UK. [4] Newcastle Fertility Centre, Centre for Life, Times Square, Newcastle upon Tyne NE1 4EP, UK. [5] Graduate School of Medicine, Osaka City University, Osaka 545-8585, Japan. [6] IVF Namba Clinic, Osaka 550-0015, Japan. [7] Laboratory for Animal Resources and Genetic Engineering, RIKEN Center for Biosystems Dynamics Research (BDR), Kobe 650-0047, Japan. ✉email: tomoya.kitajima@riken.jp

**B**ipolar spindle formation is a prerequisite for chromosome segregation. In animal somatic cells, the two centrosomes act as major microtubule nucleation sites and provide spatial cues for the establishment of spindle bipolarity. However, in oocytes of many species, including humans, the meiotic bipolar spindle forms with no centrosomes[1–7]. The acentrosomal nature of the spindle has been implicated in error-prone chromosome segregation in oocytes[2,8–10]. Live imaging analysis of human oocytes has shown that the failure to establish spindle bipolarity precedes a majority of chromosome segregation errors in meiosis I (MI)[11,12]. Chromosome segregation errors in oocytes, which are more predominant in MI than in meiosis II (MII), cause aneuploidy in the resulting eggs, which is the leading cause of pregnancy loss and several congenital disease, such as Down syndrome[13,14].

Microtubule nucleation promoted by RanGTP is a key pathway in the formation of the bipolar spindle in acentrosomal cells. This pathway is activated around chromosomes, and is required for efficient acentrosomal spindle assembly in fly, frog, mouse, and human oocytes[11,15–18]. In mouse oocytes, many acentriolar microtubule organizing centers (MTOCs), which are activated around chromosomes through well-studied pathways including the RanGTP pathway, act as major sites for microtubule nucleation[16,17,19–23]. While much is known about the structures that function as scaffolds to promote microtubule nucleation, less is known about the structures that promote spindle bipolarization. The balanced activities of microtubule regulators, such as the plus-end-directed motor Kif11 (also known as kinesin-5 and Eg5), the minus-end-directed motor HSET (kinesin-14), the bundle stabilizer HURP, and the minus-end clustering factor NuMA, that act on microtubules are critical for spindle bipolarization[17,19,21,24–27]. However, scaffolds that primarily enrich spindle bipolarization factors independently of microtubules have not been identified.

The kinetochore is a macromolecular structure that links chromosomes to microtubules through multiple protein complexes[28–30]. The primary microtubule receptor at the kinetochore is the Ndc80 complex, which is composed of Ndc80 (also known as Hec1), Nuf2, Spc24, and Spc25[31,32]. Ndc80 and Nuf2 mediate microtubule attachment through their microtubule-binding domains and are linked to the kinetochore scaffold through Spc24 and Spc25[33–35]. Several lines of evidence suggest that kinetochores contribute to spindle bipolarity. In centrosomal mitotic cells, the loss of kinetochore–microtubule attachments can cause defects in spindle bipolarity, which are pronounced when centrosomal functions are perturbed[36–39]. In mouse oocytes, mutations and knockdowns that cause defects in kinetochore–microtubule attachment, including knockdown of the Ndc80 complex, can perturb spindle bipolarity[40–43]. Based on these previous findings, it is generally thought that forces mediated by kinetochore–microtubule attachment contribute to spindle bipolarity. Whether kinetochores play a role independent of microtubule attachment in spindle bipolarization is unknown.

It is well established that kinetochores are dispensable for acentrosomal spindle formation in frog and mouse MII oocytes, because DNA-coated beads can form a bipolar spindle in these cells[44–46]. It is unknown, however, whether kinetochores are required for spindle formation in MI, which may be fundamentally different from that in MII. Consistent with this possibility, MI and MII spindles exhibit markedly different sensitivities to RanGTP inhibition in mouse oocytes[16]. Moreover, spindle formation in MI proceeds slowly (4–6 h in mice and ~16 h in humans), whereas spindle formation in MII is rapid (<1 h in mice and ~3 h in humans)[11,16,17]. Several microtubule regulators such as HSET, Kif11, Tpx2, Miss, and the phosphorylation of Tacc3 are upregulated in MII[26,27,47,48], which may contribute to rapid

spindle bipolarization. Consistent with this idea, HSET overexpression accelerates spindle bipolarization in MI[26,27]. The full repertoire of factors that contribute to the difference in the kinetics of spindle formation between MI and MII is unknown.

Here, we identify the kinetochore as a structure that promotes error-free spindle bipolarization during MI in mouse oocytes. Mouse oocytes lacking the kinetochore Ndc80 fail to form a bipolar spindle in MI, but not in MII. We find that the Ndc80 complex recruits the antiparallel microtubule crosslinker Prc1[49,50] to kinetochores, independently of its microtubule attachment, in an oocyte-specific manner. The Prc1-rich kinetochore microenvironment is required for slow-mode spindle bipolarization in MI. In contrast, MII spindles rapidly form a bipolar-shaped spindle largely independently of Ndc80, with support from the cytoplasmic environment that contain upregulated factors including Prc1. The kinetochore-dependent mode of spindle bipolarization is required for error-free chromosome segregation in MI. In contrast to mouse oocytes, human oocytes, which have been reported to undergo error-prone spindle bipolarization[11,12], exhibit little enrichment of Prc1 at kinetochores. The present findings reveal an oocyte-specific role for kinetochores in acentrosomal spindle bipolarization during MI, and provide a better understanding of how the mode of spindle bipolarization differs between MI and MII in mouse oocytes. Based on the findings in human oocytes, we discuss molecular and evolutionary insights into susceptibility to chromosome segregation errors in MI.

## Results

**Oocyte-specific deletion of the *Ndc80* gene**. To examine the role of functional kinetochores in mouse oocytes, we deleted the gene encoding Ndc80, a kinetochore component that anchors spindle microtubules[28–30]. We inserted *loxP* sites surrounding exon 2 of the *Ndc80* gene (*Ndc80*f/f) (Supplementary Fig. 1a, b). The *Ndc80*f/f mice were crossed to mice that express Cre recombinase under the control of the *Zona pellucida 3* promoter (*Zp3-Cre*), which is specifically activated during the early stages of growing oocytes[51]. Western blot analysis showed that Ndc80 was efficiently removed in fully-grown oocytes harvested from *Ndc80*f/f *Zp3-Cre* mice (Supplementary Fig. 1c). Consistent with this observation, immunostaining for Ndc80 and Nuf2 was undetectable at kinetochores in MI and MII (Supplementary Fig. 1d). These results demonstrate the efficient removal of Ndc80 in oocytes from *Ndc80*f/f *Zp3-Cre* mice (hereafter called *Ndc80*-deleted oocytes).

**Ndc80 is essential for spindle bipolarization in MI but not in MII**. We next recorded 4D image datasets of microtubule and chromosome dynamics in living oocytes throughout meiotic maturation (Fig. 1a). Analysis of the 3D-reconstructed signals of the microtubule marker EGFP-Map4[17] showed that during MI in control oocytes (*Ndc80*f/f), microtubules were nucleated upon nuclear envelope breakdown (NEBD, 0 h) and underwent a slow process of bipolarization exhibiting multiple attempts to reorganize an apolar or multipolar microtubule mass into a stable bipolar-shaped spindle (3.4 ± 1.3 h). The oocytes then underwent anaphase I, which was followed by relatively rapid reformation of a bipolar-shaped MII spindle within 0.1 ± 0.1 h after entry into MII (Fig. 1a, b). These observations were consistent with those from previous studies using wild-type oocytes[17]. In contrast, in *Ndc80*-deleted oocytes, although the total volume of MTOCs and the kinetics of microtubule nucleation upon NEBD were largely intact (Fig. 1a, Supplementary Fig. 2a, b), the microtubule mass underwent dynamic and aberrant shape changes and failed to establish a stable bipolar-shaped spindle during MI (0 of 58 oocytes) (Fig. 1a, b). Consistent with severe

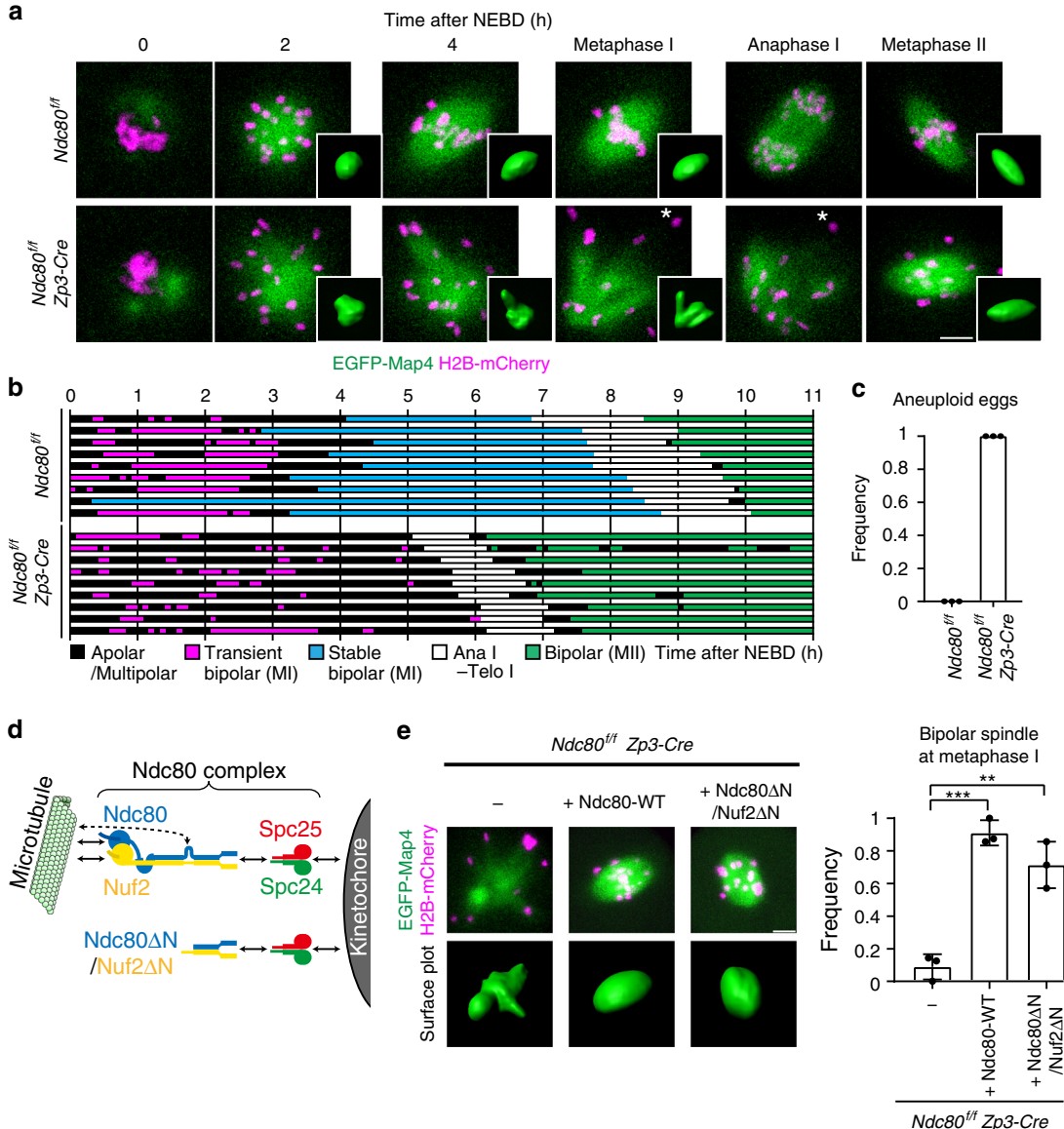

**Fig. 1 Ndc80 is essential for spindle bipolarization in MI but not in MII. a** Live imaging of *Ndc80[f/f]* (control) and *Ndc80[f/f] Zp3-Cre* (*Ndc80*-deleted) oocytes expressing EGFP-Map4 (microtubules, green) and H2B-mCherry (chromosomes, magenta). Spindle shapes were reconstructed in 3D. Asterisks show chromosomes that fell off the spindle. Anaphase I was defined based on characteristic chromosome and spindle dynamics (see Methods). Three independent experiments were performed. See also Supplementary Movie 1, 2, and 3. **b** Establishment of spindle bipolarity requires Ndc80 in MI. Spindle shapes in 3D were categorized based on the aspect ratio, surface irregularity, and stability (see Methods). Spindles that maintained a bipolar state for 1 h or longer with no collapse prior to anaphase I onset were categorized as stable bipolar spindles. **c** Ndc80 is required for preventing aneuploidy in eggs. The number of chromosomes were counted at metaphase II (*n* = 28, 28 oocytes from three independent experiments). Bars represent mean. See also Supplementary Movie 4. **d** The Ndc80 complex and Ndc80ΔN/Nuf2ΔN construct. In the Ndc80 complex, Ndc80 and Nuf2 directly bind to microtubules via the head domains at their N-termini, and to Spc24 and Spc25 at their C-termini. Ndc80 contains a loop domain in the central region, which recruits microtubule-binding proteins. Ndc80ΔN (a.a. 461–642) and Nuf2ΔN (a.a. 276–463) contain neither the head nor loop domain but retain the Spc24- and Spc25-binding domains. **e** Ndc80ΔN/Nuf2ΔN rescues spindle defects. *Ndc80[f/f] Zp3-Cre* oocytes coexpressing Ndc80ΔN and Nuf2ΔN (Ndc80ΔN/Nuf2ΔN) and those expressing full-length Ndc80 (Ndc80-WT) were monitored for spindle formation. EGFP-Map4 (microtubules, green) and H2B-mCherry (chromosomes, magenta) signals at metaphase I (5.5 h after NEBD) are shown. Spindle shapes were reconstructed in 3D at metaphase I (5.5 h after NEBD) and categorized based on the aspect ratio and surface irregularity (*n* = 23, 23, 21 oocytes from three independent experiments). Mean +/− SD are presented. **p = 0.0026; ***p = 0.0002; by two-tailed unpaired Student's *t*-test. See also Supplementary Movie 5. Scale bars, 10 μm.

kinetochore–microtubule attachment defects, some chromosomes fell off the microtubule mass (Fig. 1a) The *Ndc80*-deleted oocytes then underwent aberrant anaphase I accompanied by degradation of securin (Supplementary Fig. 2c), a reporter for anaphase entry[52,53], consistent with spindle checkpoint defects (Supplementary Fig. 2d)[42,54,55]. This aberrant anaphase resulted in egg aneuploidies, including those derived from chromosome

nondisjunction (Fig. 1c, Supplementary Figs. 2e and 3). Intriguingly, despite the severe spindle defects in MI, the *Ndc80*-deleted oocytes frequently established a stable bipolar-shaped spindle in MII (38 of 56 oocytes, Fig. 1a, b). These bipolar-shaped spindles exhibited massive chromosome misalignment (Fig. 1a, metaphase II), suggesting that kinetochore–microtubule attachment was still defective. Taken together, these results

demonstrate that Ndc80 is essential for spindle bipolarization in MI. MII oocytes, however, have the capacity to form a bipolar-shaped spindle in an Ndc80-independent manner.

**Cyclin B2 is intact in *Ndc80*-deleted oocytes**. A previous study showed that introduction of morpholinos against *Ndc80* into germinal vesicle (GV)-stage (prophase I) oocytes resulted in a delay in NEBD and defects in the early stages of spindle assembly, both of which were attributed to the loss of Ndc80-mediated stabilization of cyclin B2[43]. In *Ndc80^{f/f} Zp3-Cre* oocytes, however, a delay in NEBD was not observed (Supplementary Fig. 4a). Western blotting of *Ndc80^{f/f} Zp3-Cre* oocytes showed intact levels of cyclin B2 (Supplementary Fig. 4b). Moreover, in contrast to *Ndc80* morpholino-injected oocytes[43], overexpression of cyclin B2 failed to rescue spindle defects in *Ndc80^{f/f} Zp3-Cre* oocytes (Supplementary Fig. 4c). Thus, the spindle bipolarization defects observed in *Ndc80^{f/f} Zp3-Cre* oocytes are unlikely to be due to the loss of Ndc80-mediated cyclin B2 stabilization.

**The C-terminal domains of Ndc80 and Nuf2 promote spindle bipolarization in MI**. We next tested the possibility that Ndc80 promotes spindle bipolarization via kinetochore–microtubule attachments. The Ndc80 complex directly binds to microtubules via the N-terminal head domains of Ndc80 and Nuf2, and recruits microtubule-binding proteins via the central loop domain of Ndc80 (Fig. 1d)[28–30]. Unexpectedly, we found that coexpression of the C-terminal fragments of Ndc80 and Nuf2 (hereafter called Ndc80ΔN/Nuf2ΔN), which contained neither the head nor loop domains (Fig. 1d), significantly rescued spindle bipolarization defects in *Ndc80*-deleted oocytes (Fig. 1e). It is unlikely that Ndc80ΔN cooperated with endogenous Nuf2 because the expression of Ndc80ΔN alone did not facilitate localization to kinetochores or rescue spindle bipolarization (Supplementary Fig. 5a, b). The bipolar-shaped spindles rescued by Ndc80ΔN/Nuf2ΔN still exhibited extensive chromosome misalignment (Fig. 1e) and failed to establish kinetochore–microtubule attachment (Supplementary Fig. 5c). Thus, these results suggest that the C-terminal domains of Ndc80 and Nuf2 promote spindle bipolarization through a pathway independent of kinetochore–microtubule attachments.

**Ndc80/Nuf2 interacts with the antiparallel microtubule crosslinker Prc1 at kinetochores**. To explore how the C-terminal domains of Ndc80 and Nuf2 promote spindle bipolarization, we employed a yeast two-hybrid screening using Ndc80ΔN/Nuf2ΔN as a bait against a cDNA library of GV-stage mouse oocytes. This screen identified Prc1 (Fig. 2a), a crosslinker of antiparallel microtubules[49,50]. Previous studies showed that in centrosomal mitosis of cultured cells, Prc1 is undetectable at kinetochores and dispensable for bipolar spindle formation[56–58]. We found that, in mouse oocytes, Prc1 localized to kinetochores (Fig. 2b and Supplementary Fig. 6a), unlike in centrosomal mitotic cells (Fig. 2c and Supplementary Fig. 6b) and in centrosomal meiotic spermatocytes (Fig. 2d and Supplementary Fig. 6c). In oocytes, before spindle bipolarization (prometaphase I), Prc1 localized almost exclusively at kinetochores (Fig. 2b). A closer examination showed that Prc1 colocalized with Ndc80 at the outer kinetochore (Fig. 2e). Kinetochore localization was confirmed with fluorescently tagged Prc1 (Supplementary Fig. 6d–f). As the spindle became bipolarized, Prc1 localized not only at kinetochores but also on the spindle (Fig. 2b, metaphase I). During anaphase I, Prc1 accumulated at the spindle midzone (Fig. 2b), which was consistent with observations using mitotic cells (Supplementary Fig. 6b)[56–58]. During metaphase II, Prc1 was detected at kinetochores and strongly enriched along microtubules in the central region of the spindle (Fig. 2b and Supplementary Fig. 6g).

Kinetochore Prc1 levels at prometaphase I were not reduced by treatment with either nocodazole, a drug that induces microtubule depolymerization (Fig. 2f), or monastrol, a drug that blocks spindle bipolarization by inhibiting kinesin-5 Kif11 (Fig. 2f). These results suggest that oocytes, but not other cell types, strongly enrich Prc1 at kinetochores via the Ndc80/Nuf2 complex prior to and independently of kinetochore–microtubule attachments and spindle bipolarization.

**Prc1 is required to promote spindle bipolarization in MI**. To address whether Prc1 is required for spindle bipolarization in oocytes, we depleted Prc1 in oocytes by RNAi. RNAi significantly reduced the total and kinetochore levels of Prc1 (Supplementary Fig. 7a, b). Live imaging revealed that Prc1 depletion significantly delayed spindle bipolarization (Fig. 3a, b), increased misaligned chromosomes (Fig. 3a, c), and delayed anaphase I onset in a spindle checkpoint-dependent manner (Fig. 3a, d). After entry into anaphase I, the oocytes failed to complete cytokinesis (Supplementary Fig. 7e), which was consistent with the phenotypes observed in Prc1-depleted mitotic cells[56–58]. Introduction of an RNAi-resistant form of Prc1 rescued all of these defects (Supplementary Fig. 7c–e). These data show that Prc1 is required for timely spindle bipolarization, which is critical for chromosome segregation during MI in oocytes.

**Ndc80/Nuf2 promotes spindle bipolarization by recruiting Prc1 at kinetochores**. We then examined the possibility that Ndc80/Nuf2 promotes spindle bipolarization through Prc1 at kinetochores in MI. *Ndc80*-deleted oocytes exhibited nearly undetectable levels of kinetochore Prc1, which was recovered by the expression of full-length Ndc80 or Ndc80ΔN/Nuf2ΔN (Fig. 4a). In contrast, the localizations of HSET, Kif11, HURP, and Aurora A, microtubule regulators that contribute to spindle bipolarization[19,21,23,26], were not directly affected by *Ndc80* deletion (Supplementary Fig. 8). Ndc80-bound beads were co-associated with Prc1 in oocytes, suggesting their physical interaction (Fig. 4b). Mutagenesis of Ndc80ΔN/Nuf2ΔN revealed that the mutant form Ndc80ΔN-4A/Nuf2ΔN, which did not localize to kinetochores (Supplementary Fig. 9a, indicated as 'NNΔN-4A') but retained the capacity to interact with Prc1 (Supplementary Fig. 9b), failed to rescue spindle defects in *Ndc80*-deleted oocytes (Fig. 4c, indicated as 'NNΔN-4A'). However, this mutant form rescued spindle defects when tethered to kinetochores by fusion to the kinetochore-targeting domains of Spc25 and Spc24[35] (Fig. 4c; indicated as 'NNΔN-4A tethered to KT'). Moreover, kinetochore-tethered Ndc80/Nuf2 constructs required Prc1-recruiting domains to rescue the spindle defects (Supplementary Fig. 9c, d; indicated as 'NNΔC tethered at KT'). Accordingly, when Prc1 was depleted, Ndc80ΔN/Nuf2ΔN did not significantly rescue spindle bipolarization defects in *Ndc80*-deleted oocytes (Fig. 4d, e). These data suggest that Ndc80/Nuf2 promotes spindle bipolarization by recruiting Prc1 to kinetochores.

**Ndc80/Nuf2 concentrates dynamic Prc1 at kinetochores for spindle bipolarization**. Fluorescence recovery after photobleaching (FRAP) analysis at prometaphase I revealed that kinetochore-enriched Prc1 exhibited a rapid turnover ($t^{1/2} = 17.7 \pm 4.8$ s) compared to Ndc80 (Fig. 5a). At metaphase I, Prc1 was also enriched in the central region of the bipolar spindle, particularly along microtubule bundles, including those closely associated with kinetochores (Fig. 5b). The Prc1 enrichment on kinetochore-proximal microtubule bundles was Kif11-dependent (Fig. 5b). These observations suggest that Prc1 undergoes dynamic exchanges at kinetochores, and following spindle bipolarization, it also marks kinetochore-proximal microtubule

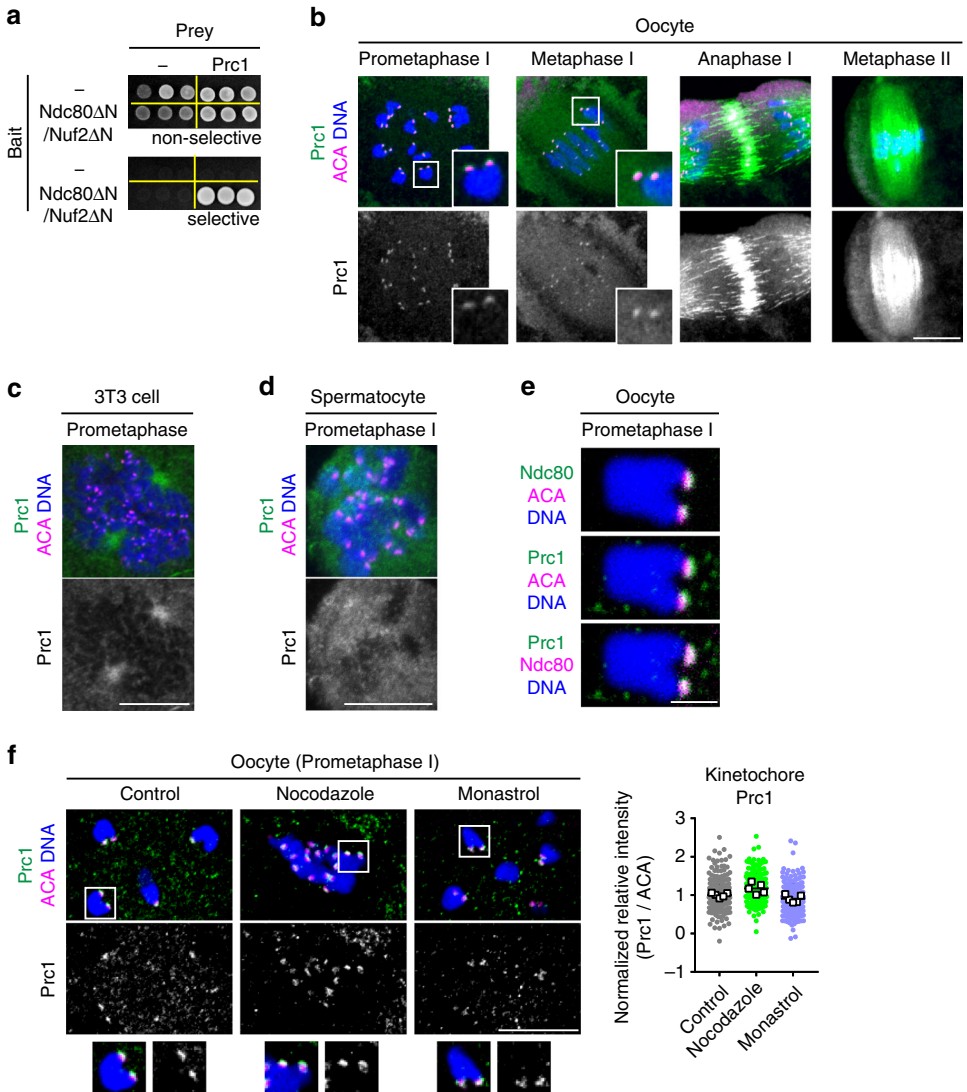

**Fig. 2 Ndc80/Nuf2 interacts with the antiparallel microtubule crosslinker Prc1 at kinetochores. a** Prc1 can interact with Ndc80ΔN/Nuf2ΔN. Yeast two-hybrid assay using Ndc80ΔN fused with a DNA-binding domain, Nuf2ΔN, and Prc1 fused with a transcription activation domain. Selective (−His, −Ade) and nonselective (+His, +Ade) plates were used. **b** Prc1 localizes to kinetochores. Z-projection images of oocytes stained for Prc1 (green), kinetochores (ACA, magenta), and DNA (Hoechst33342, blue) are shown. Prc1 signals at kinetochores are magnified. One of the z-slice images at metaphase II is shown in Supplementary Fig. 6g. More than three independent experiments were performed. **c** Prc1 is not enriched at kinetochores at mitotic prometaphase in centrosomal cultured cells. NIH3T3 cells were stained for Prc1 (green), kinetochores (ACA, magenta), and DNA (Hoechst33342, blue). Three independent experiments were performed. **d** Prc1 is not enriched at kinetochores at meiotic prometaphase I in centrosomal spermatocytes. Spermatocytes were stained for Prc1 (green), kinetochores (ACA, magenta), and DNA (Hoechst33342, blue). Three independent experiments were performed. **e** Prc1 colocalizes with Ndc80. Oocytes expressing Ndc80-sfGFP at prometaphase I (2 h after NEBD) were immunostained with anti-GFP, anti-Prc1, and ACA antibodies. DNA is counterstained with Hoechst33342 (blue). Magnified images show that Prc1 is located closer to Ndc80 than to ACA. Three independent experiments were performed. **f** Prc1 localizes to kinetochores independently of microtubule attachment and spindle bipolarization. Oocytes were treated with nocodazole or monastrol at 1.5 h after NEBD, incubated for 30 min, and immunostained for Prc1 (green), kinetochores (ACA, magenta), and DNA (Hoechst33342, blue). Prc1 signals at kinetochores are magnified. The kinetochore ratios of the Prc1 signals to the ACA signal are shown. Spots and squares correspond to individual kinetochores and oocytes, respectively ($n = 200, 200, 198$ kinetochores from 5, 5, 5 oocytes). Three independent experiments were performed. Mean +/− SD are presented. Scale bars; 10 μm (**b**), (**c**), (**d**), and (**f**); 2 μm (**e**).

bundles. Notably, we found that Prc1 overexpression significantly rescued the spindle defects in *Ndc80*-deleted oocytes (Fig. 5c), which indicated that a global increase in Prc1 levels partially overcame the need for local enrichment. In contrast, neither of the overexpression of HSET or Kif11 significantly rescued the spindle defects in *Ndc80*-deleted oocytes (Fig. 5c). HURP overexpression slightly, although not significantly, rescued the spindle defects (Fig. 5c), which may suggest functional overlap between HURP and Prc1. Taken together, these results suggest that the

Ndc80/Nuf2 complex promotes spindle bipolarization by concentrating a dynamic pool of Prc1 at kinetochores.

**Limited expression of Prc1 allows the MI-specific mode of spindle bipolarization and ensures chromosome segregation fidelity.** The above results show that spindle bipolarization depends on Ndc80 functions in MI, whereas Ndc80-independent pathways support spindle bipolarization in MII. To determine whether cytoplasmic factors in MII are sufficient to support

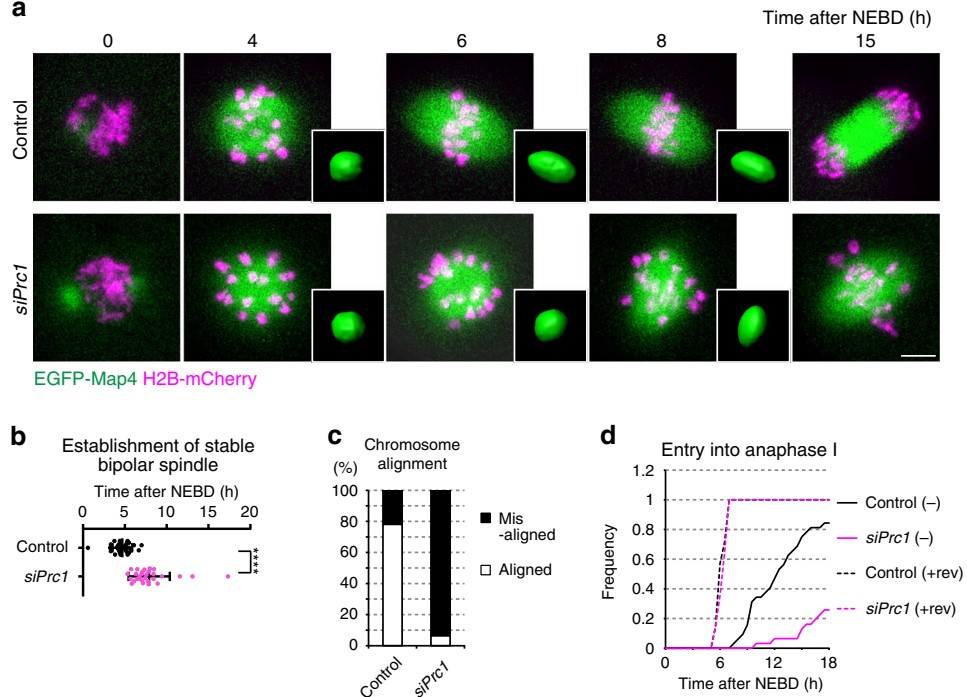

**Fig. 3 Prc1 is required for timely spindle bipolarization in MI. a** Live imaging of oocytes after Prc1 RNAi. *Ndc80*[f/f] oocytes labeled with EGFP-Map4 (microtubules, green) and H2B-mCherry (chromosomes, magenta) were used. Spindle shapes were reconstructed in 3D. Four independent experiments were performed. Scale bar, 10 μm. See also Supplementary Movie 6. **b** Prc1 depletion delays spindle bipolarization. Spindle shapes in 3D were categorized based on the aspect ratio, surface irregularity, and stability (see Methods). The plot shows the time at which a stable bipolar spindle was established ($n =$ 32, 29 oocytes from four independent experiments). Mean $+/-$ SD are presented. ****$p < 0.0001$ ($p = 2.4E-08$) by two-tailed unpaired Student's *t*-test. **c** Prc1 depletion increases chromosome misalignment. Oocytes were categorized into 'aligned' when all chromosomes located at the middle half of the spindle ($n =$ 32, 31 oocytes from four independent experiments). **d** Prc1 depletion causes a delay in anaphase I onset. Oocytes were incubated with or without the checkpoint inhibitor reversine (rev). The percentages of oocytes that underwent anaphase I onset were plotted ($n =$ 32, 31, 14, 14 oocytes from at least three independent experiments).

Ndc80-independent spindle bipolarization, we fused *Ndc80*-deleted oocytes in MI to those in MII and monitored spindle formation. We found that the fused oocytes successfully formed a bipolar-shaped spindle around MI chromosomes (Fig. 6a, 10 of 10 oocytes). Thus, factors provided by the MII cytoplasm support Ndc80-independent spindle bipolarization.

We hypothesized that Prc1 is one of the cytoplasmic factors that contribute to Ndc80-independent spindle bipolarization in MII, because (1) Prc1 overexpression attenuated spindle bipolarization defects in MI of *Ndc80*-deleted oocytes (Fig. 5c) and (2) Prc1 levels along spindle microtubules were much higher in MII than in MI (Fig. 2b and Supplementary Fig. 10a). Consistent with these findings, the total levels of Prc1 were upregulated in MII compared with those in MI (Fig. 6b). These observations prompted us to examine whether an artificial increase in Prc1 expression during MI to a level comparable to that in MII (Fig. 6b) is sufficient to recapitulate MII-like spindle bipolarization. Live imaging showed that while control oocytes underwent a lengthy phase with multiple attempts to reorganize an apolar or multipolar microtubule mass into a bipolar-shaped spindle during MI, oocytes expressing increased levels of Prc1 skipped the lengthy phase and rapidly established a bipolar-shaped spindle in MI (Fig. 6c–e). Moreover, the length/width ratios of the MI spindles that formed in oocytes expressing increased levels of Prc1 resembled those of normal MII spindles (Fig. 6f). Thus, oocytes expressing increased levels of Prc1 lack the kinetic and morphogenetic features of spindle bipolarization in MI.

Using this experimental system, we asked whether the MI-specific mode of spindle bipolarization is critical for chromosome segregation fidelity. When the kinetic and morphogenetic features of spindle bipolarization were altered by expression of increased levels of Prc1 in MI, oocytes exhibited a significant decrease in stable kinetochore–microtubule attachment (Supplementary Fig. 10b), a spindle checkpoint-dependent delay in anaphase I onset (Supplementary Fig. 10c), and production of aneuploid eggs (Fig. 6g). These results demonstrate that the restricted level of Prc1 is critical for MI-specific, slow-mode spindle bipolarization, and for error-free chromosome segregation.

**Human oocytes exhibit little enrichment of Prc1 at kinetochores.** Given the importance of the kinetochore Ndc80/Nuf2–Prc1 pathway for error-free spindle bipolarization in mouse oocytes, we examined whether this pathway exists in human oocytes, where spindle bipolarization has been reported to be error prone[11,12]. Human Ndc80 interacted with human Prc1 in a two-hybrid assay (Supplementary Fig. 11a). Moreover, human Ndc80 strongly enriched human Prc1 at kinetochores when expressed in mouse oocytes (Supplementary Fig. 11b). Unexpectedly, however, in human oocytes, Prc1 did not exhibit detectable enrichment at kinetochores. Prc1 non-uniformly localized along spindle microtubules, with only occasional enrichment on fibers closely associated with kinetochores (Fig. 7a and Supplementary Fig. 11c). Human oocytes treated with colchicine, which depolymerizes microtubules, exhibited prominent kinetochore localization of Ndc80 but not of Prc1 (Fig. 7b). Importantly, kinetochore Prc1 was consistently undetectable in oocytes of young women (<30 years old) with no history of

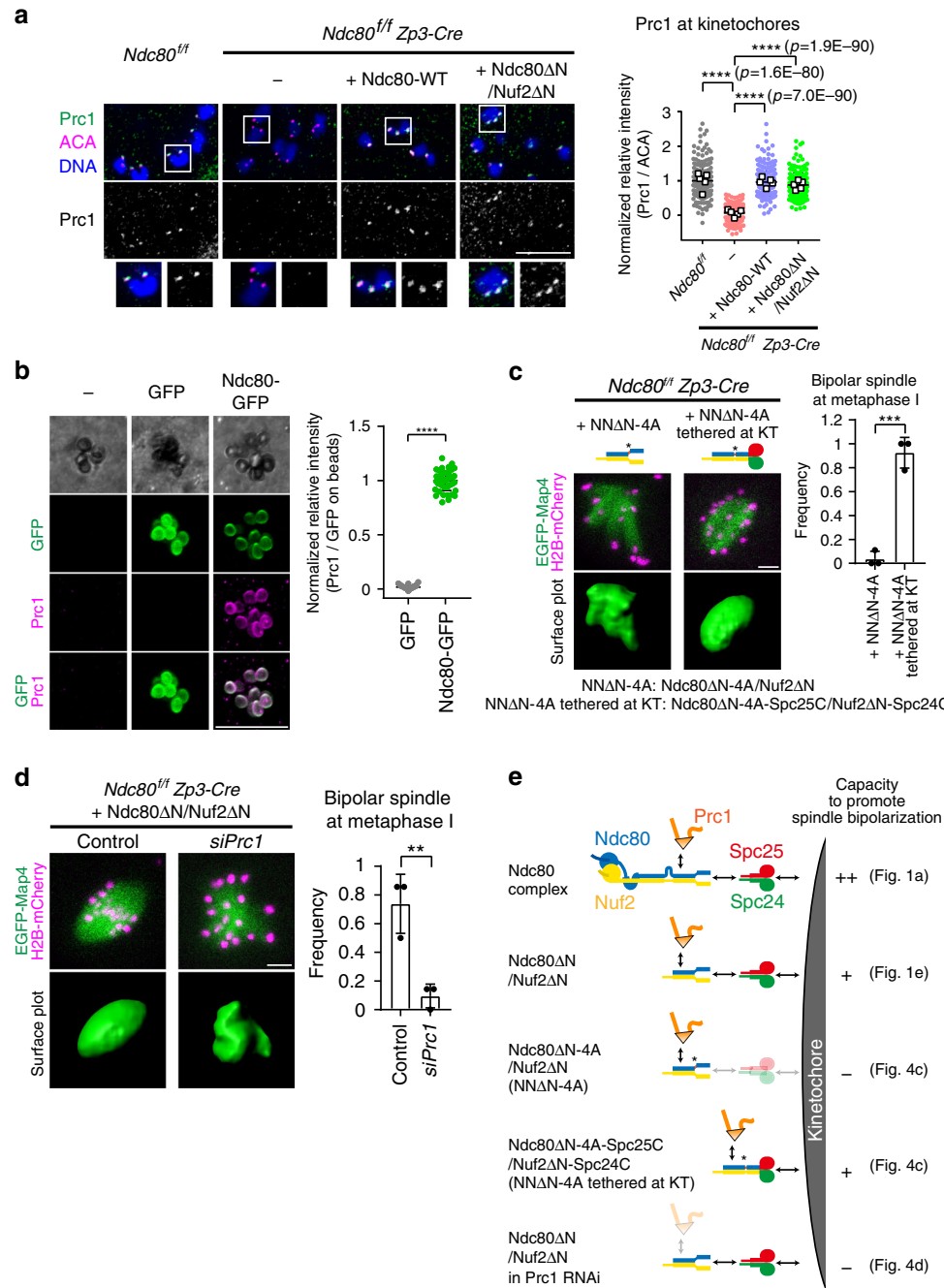

infertility (Fig. 7a). These results suggest that human oocytes are inherently less capable of enriching Prc1 at kinetochores than mouse oocytes.

## Discussion

Spindle bipolarization in acentrosomal oocytes is essential for preventing aneuploidy in eggs. The results presented here demonstrate that kinetochores are required for acentrosomal spindle bipolarization during MI in mouse oocytes, and that the modes of spindle bipolarization differ fundamentally between MI and MII (Fig. 8). Spindle bipolarization in MI proceeds slowly and depends on a kinetochore microenvironment that concentrates the antiparallel microtubule crosslinker Prc1 via its recruiter Ndc80 complex. In contrast, spindle bipolarization in MII proceeds rapidly and does not require kinetochores, as it is

supported by a cytoplasmic environment that contains upregulated factors including Prc1. The kinetochore-dependent mode of spindle bipolarization in MI is essential for error-free chromosome segregation and thereby for preventing aneuploidy in mouse eggs.

Our finding that kinetochore-dependent spindle bipolarization occurs in mouse MI oocytes suggests the importance of spatial control of microtubule regulation in acentrosomal spindle assembly. It is possible that kinetochore-based control of spindle bipolarization enables functional cooperation between kinetochore–microtubule attachment and antiparallel microtubule sorting. Expression of Ndc80ΔN/Nuf2ΔN, which rescued kinetochore recruitment of Prc1 (Fig. 4a), significantly but only partially rescued spindle bipolarization defects in Ndc80-deleted oocytes (Fig. 1e), suggesting that both the Prc1-recruiting and

**Fig. 4 Ndc80/Nuf2 promotes spindle bipolarization by recruiting Prc1 at kinetochores. a** Ndc80/Nuf2 recruits Prc1 at kinetochores. *Ndc80$^{f/f}$* (control) and *Ndc80$^{f/f}$ Zp3-Cre* (*Ndc80*-deleted) oocytes expressing full-length Ndc80 (Ndc80-WT) or Ndc80ΔN/Nuf2ΔN were immunostained for Prc1 (green), kinetochores (ACA, magenta), and DNA (Hoechst33342, blue). Kinetochores are magnified. In the plot, spots and squares correspond to individual kinetochores and oocytes, respectively (n = 200, 198, 200, 200 kinetochores from 5, 5, 5, 5 oocytes). ****p < 0.0001 (exact values are shown in the panel) by two-tailed unpaired Student's *t*-test. **b** Ndc80-bound beads associate with Prc1. Oocytes expressing mEGFP or Ndc80-mEGFP (green) were microinjected with anti-GFP beads, fixed at prometaphase I, and immunostained for Prc1 (magenta) (see Methods). In the plot, spots correspond to individual beads (n = 50, 48 beads from 5, 5 oocytes). ****p < 0.0001 (p = 4.7E−87) by two-tailed unpaired Student's *t*-test. **c** Ndc80/Nuf2–Prc1 acts at kinetochores. *Ndc80$^{f/f}$ Zp3-Cre* oocytes expressing Ndc80ΔN-4A/Nuf2ΔN (Y564A, Q565A, L566A, and T567A mutations; indicated as 'NNΔN-4A') or its Spc-fused form Ndc80ΔN-4A-Spc25C/Nuf2ΔN-Spc24C (indicated as 'NNΔN-4A tethered at KTs') were monitored. Spc25C (a.a. 120–226) and Spc24C (a.a. 122–201) are kinetochore-targeting domains. EGFP-Map4 (microtubules, green) and H2B-mCherry (chromosomes, magenta) at metaphase I (5.5 h after NEBD) are shown. Spindles were reconstructed in 3D and categorized based on the aspect ratio and surface irregularity. The frequency of oocytes that exhibited a bipolar-shaped spindle is shown (n = 19, 18 oocytes from three independent experiments). ***p = 0.0004 by two-tailed unpaired Student's *t*-test. **d** Ndc80ΔN/Nuf2ΔN requires Prc1 to promote spindle bipolarization. Prc1 was depleted by RNAi in *Ndc80$^{f/f}$ Zp3-Cre* oocytes, and Ndc80ΔN/Nuf2ΔN was expressed. EGFP-Map4 (microtubules, green) and H2B-mCherry (chromosomes, magenta) at metaphase I (8 h after NEBD) are shown. Spindle shapes were reconstructed in 3D and categorized based on the aspect ratio and surface irregularity. The frequency of oocytes that exhibited a bipolar-shaped spindle is shown (n = 27, 27 oocytes from three independent experiments). **p = 0.0021 by two-tailed unpaired Student's *t*-test. **e** Summary of Ndc80/Nuf2 constructs and their capacities to rescue spindle bipolarization. Scale bars, 10 μm. Mean +/− SD are presented in **a–d**.

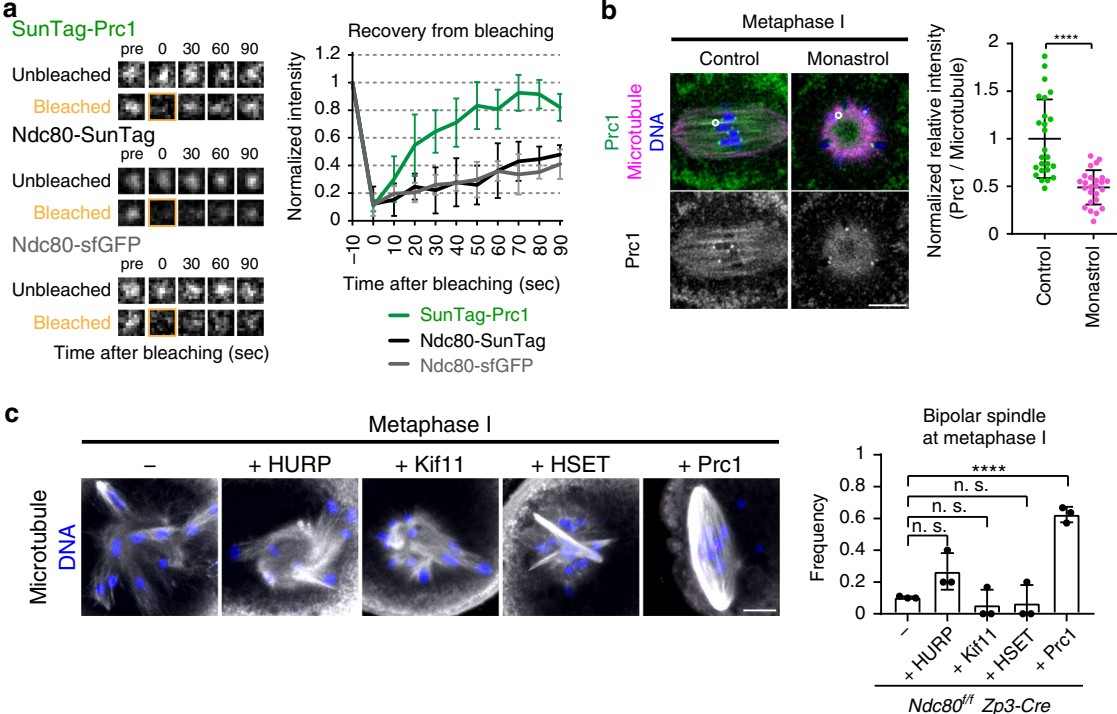

**Fig. 5 Ndc80/Nuf2 concentrates dynamic Prc1 at kinetochores for spindle bipolarization. a** Dynamic exchange of Prc1 at kinetochores. In oocytes at prometaphase I (1.5 h after NEBD), SunTag-Prc1 (24xGCN4-Prc1 coexpressed with scFv-sfGFP, green) signals at kinetochores were bleached, and the recovery was monitored (n = 5, 5, 5 kinetochores). Note that the recovery curve of SunTag-Prc1 signals indicates the turnover of Prc1 at kinetochores rather than the turnover of scFv on 24xGCN4 in this time range, which was confirmed by Ndc80-SunTag (Ndc80-24xGCN4 coexpressed with scFv-sfGFP) exhibiting similar recovery curves to Ndc80-sfGFP. See also Supplementary Movie 7. **b** Kif11-dependent Prc1 enrichment along kinetochore-proximal microtubules of the bipolar spindle. Control or monastrol-treated oocytes at metaphase I (4–6 h after NEBD) were stained for Prc1 (green), microtubules (magenta), and DNA (Hoechst33342, blue). The oocytes were treated with a cold buffer for 1 min before fixation to facilitate antibody penetration into the spindle. Prc1 signals along kinetochore-proximal microtubule bundles were measured, and their ratio to microtubule signals was calculated (n = 25, 25 locations from 5, 5 oocytes. Three independent experiments were performed). ****p < 0.0001 (p = 8.1E−07) by two-tailed unpaired Student's *t*-test. **c** Prc1 overexpression rescues spindle defects in *Ndc80*-deleted oocytes. *Ndc80$^{f/f}$ Zp3-Cre* oocytes overexpressing mEGFP-HURP, mEGFP-Kif11, mNeonGreen-HSET, or mNeonGreen-Prc1 were immunostained at metaphase I (5.5 h after NEBD). Spindle shapes were reconstructed in 3D and categorized (n = 29, 18, 18, 18, 24 oocytes from three independent experiments). ****p < 0.0001 (p = 5.1E−05) by two-tailed unpaired Student's *t*-test. n.s., not significant. Scale bars, 10 μm. Mean +/− SD are presented in **a–c**.

microtubule-binding domains of the Ndc80 complex contribute to spindle bipolarization. The microtubule-binding domains appear to increase the local concentration of microtubules around kinetochores, which may locally facilitate Prc1-mediated marking of antiparallel microtubule overlaps. Another possibility is that kinetochore enrichment of Prc1-dependent antiparallel

microtubule crosslinking helps kinetochores bind laterally to microtubule bundles. Such lateral kinetochore–microtubule attachments are predominantly observed during spindle bipolarization and develop into stable end-on attachments by the onset of anaphase I in mouse oocytes[59–63]. In either case, kinetochores act as a scaffold to increase the local concentration of

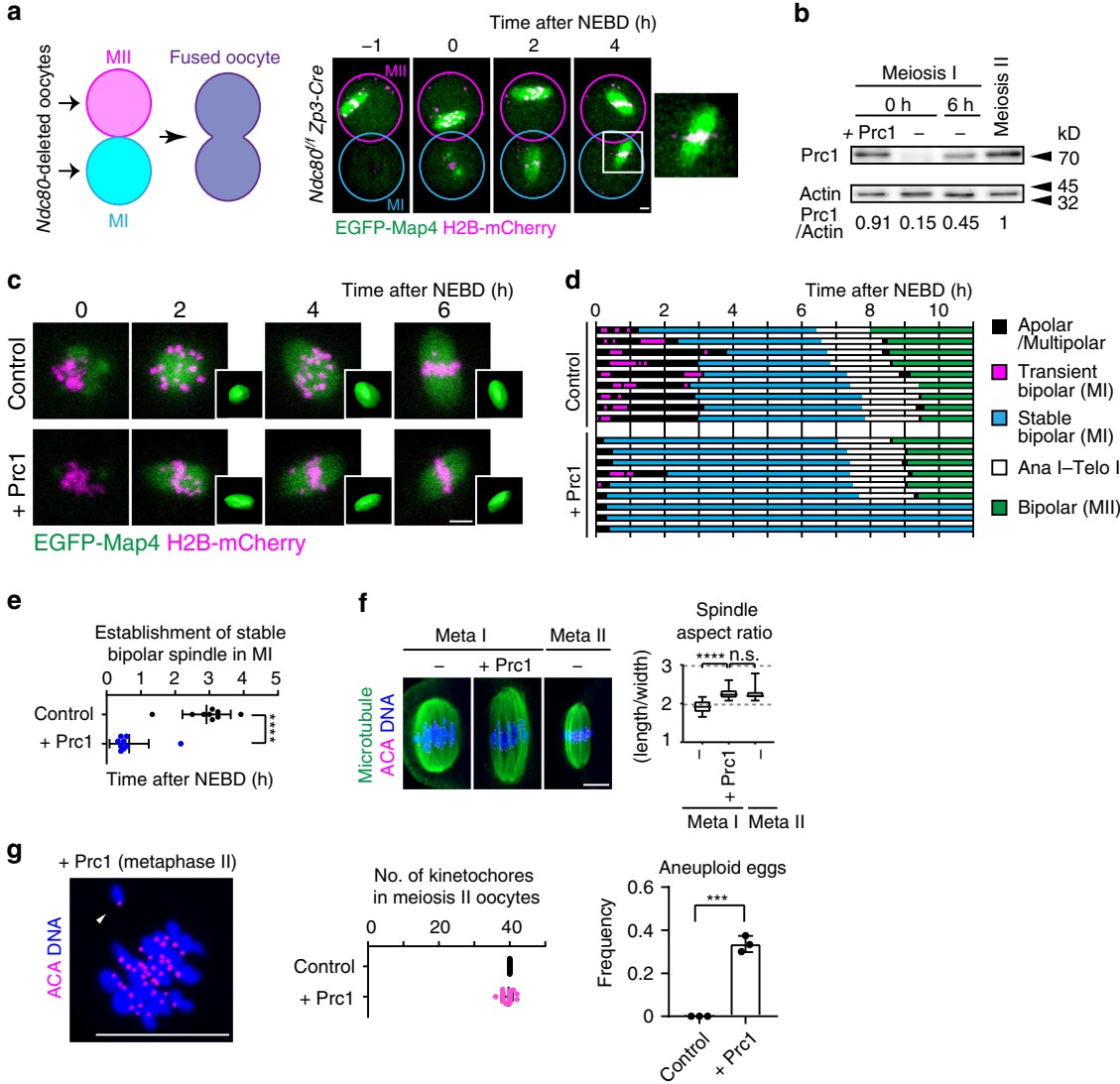

**Fig. 6 Cytoplasm containing upregulated Prc1 facilitates kinetochore-independent pathways for spindle bipolarization in MII. a** MII cytoplasm supports Ndc80-independent spindle bipolarization. An *Ndc80^{f/f} Zp3-Cre* oocyte in metaphase II (MII) and an *Ndc80^{f/f} Zp3-Cre* oocyte at the GV stage (MI) were fused. EGFP-Map4 (microtubules, green) and H2B-mCherry (chromosomes, magenta) are shown. Bipolar spindle formation around MI chromosomes (magnified) was observed in 10 of 10 oocytes from three independent experiments. Time after NEBD of the MI nucleus (h). See also Supplementary Movie 8. **b** Upregulation of Prc1 in MII and artificial increase in Prc1 levels in MI. Western blotting of oocytes at MI (0 and 6 h after NEBD), and at MII (16 h). The Prc1 level at 0 h was artificially increased (+Prc1). Fifty oocytes were used in each sample. A full scan image is provided in the Source Data file. **c** Live imaging of oocytes expressing increased Prc1. EGFP-Map4 (microtubules, green) and H2B-mCherry (chromosomes, magenta) were monitored. Spindles were reconstructed in 3D. Time after NEBD. Three independent experiments were performed. See also Supplementary Movie 9. **d** Increased Prc1 accelerates spindle bipolarization. Spindles reconstructed in 3D were categorized (see Methods). **e** Acceleration of bipolar spindle establishment. Time at which a stable bipolar spindle was established are plotted (n = 9, 9 oocytes). The mean +/− SD are shown. ****p = 1.3E−06. **f** MII-like spindle shapes. Oocytes were immunostained for microtubules (green), kinetochores (ACA, magenta) and DNA (Hoechst33342, blue) at metaphase I (6 h after NEBD) and metaphase II (16 h after NEBD). Aspect ratios (length/width) were plotted (n = 19, 8, 11 oocytes). Boxes show the 25th to 75th percentiles and whiskers show the 1st to 99th percentiles. ****p = 2.54E−05. n.s., not significant. **g** Increased Prc1 increases aneuploidy in eggs. Oocytes at metaphase II were immunostained for kinetochores (ACA, magenta) and counterstained for DNA (Hoechst33342, blue). Arrowheads indicate a misaligned chromatid in an aneuploid egg. The number of kinetochores were counted (n = 33, 40 oocytes from three independent experiments). Mean +/− SD are presented. ***p = 0.0001. Statistical values were obtained from two-tailed unpaired Student's test. Scale bars, 10 μm.

Prc1 and possibly other microtubule regulators to promote spindle bipolarization. The limited amount of Prc1 in MI allows it to initially localize almost exclusively at kinetochores and restricts the speed of spindle bipolarization. This idea is supported by the fact that artificially increased Prc1 localizes not only at kinetochores but also spindle microtubules (Supplementary Fig. 12) and accelerates spindle bipolarization (Fig. 6). In contrast, MII spindles form rapidly and are largely independent of functional kinetochores, with support from the cytoplasm that contain upregulated spindle bipolarizers including Prc1. These findings, along with previous findings that several microtubule regulators are upregulated in MII[26,27,47,48], at least partly explain why spindle formation in MI takes longer than that in MII (4–6 h in MI vs. <1 h in MII in mouse oocytes; ~16 h in MI vs. ~3 h in MII in human oocytes)[11,16,17]. Our observation that Ndc80-independent spindle bipolarization occurs during MII in mouse

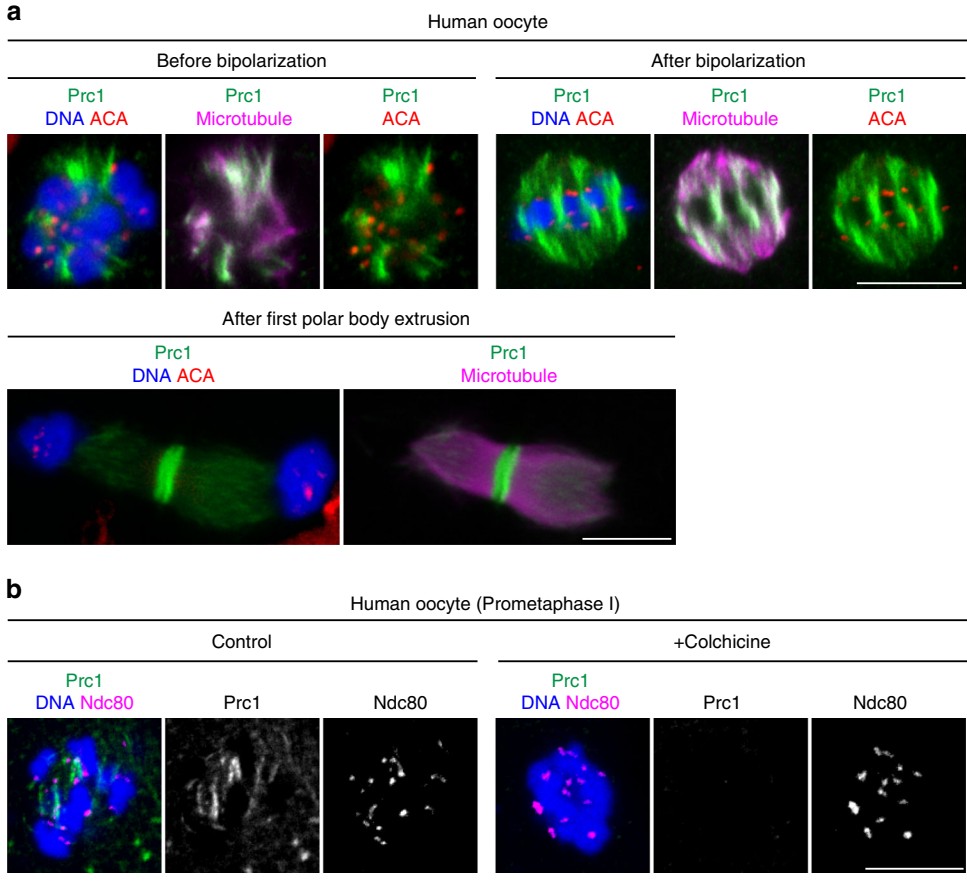

**Fig. 7 Human oocytes do not exhibit detectable enrichment of Prc1 at kinetochores. a** Prc1 localizes along spindle microtubules but not at kinetochores in human oocytes. MI oocytes of young women with no history of infertility were immunostained for Prc1 (green), kinetochores (ACA, red), microtubules (magenta) and DNA (DAPI, blue). Stages are categorized based on chromosome arrangement. Prc1 is detected at the spindle midzone after anaphase I. Oocytes from three women were examined. **b** Prc1 is not detectable at kinetochores. Human oocytes 5 h post NEBD were treated with the microtubule depolymerizer colchicine and immunostained for Prc1 (green), Ndc80 (red), and DNA (Hoechst33342, blue). Oocytes from more than three women were examined. Scale bars, 10 μm.

oocytes is consistent with previous findings that bipolar spindle formation does not require kinetochores in frog or mouse MII oocytes[44–46].

Induction of an MII-like, kinetochore-independent pathway for spindle bipolarization in MI oocytes delays the formation of kinetochore–microtubule attachments and results in error-prone chromosome segregation. This is in contrast to normal MII oocytes where kinetochore–microtubule attachments are rapidly established and undergo error-free chromosome segregation. These observations suggest that the MII-like, kinetochore-independent pathway for spindle bipolarization is not optimal for the establishment of kinetochore–microtubule attachments on MI chromosomes. These notions are consistent with previous reports that accelerated spindle bipolarization, which can be induced by increased levels of HSET expression, causes chromosome alignment defects[26,27]. Kinetochore–microtubule attachments in MI are inherently unstable due to MI-specific chromosomal properties and thus require a considerably long time to stabilize[62,63]. The kinetochore-dependent mode of spindle bipolarization in mouse MI oocytes may have evolved as an adaptation to difficulties in establishing kinetochore–microtubule attachments with MI-specific chromosomal properties.

Spindle bipolarization errors are rarely observed in mouse oocytes while predominantly observed in human oocytes[11]. The error-prone process of spindle bipolarization likely predisposes human oocytes of all ages to chromosome segregation errors. We found that human oocytes, even those of young women with no history of infertility (Fig. 7), carry Prc1-less kinetochores, although our data do not exclude the possibility that an undetectable pool of Prc1 at kinetochores contributes to spindle bipolarization. These findings may give a molecular explanation to why human but not mouse oocytes undergo error-prone spindle bipolarization. Human Ndc80 enriches human Prc1 at kinetochores when expressed in mouse oocytes (Supplementary Fig. 11b), which suggests that mouse oocytes contain factors that enhance Ndc80-mediated Prc1 enrichment and thereby enable error-free chromosome segregation during MI. Future studies are needed to explore mouse-specific factors that enhance Prc1 enrichment at kinetochores.

In animal somatic cells, centrosomes provide spatial cues for spindle bipolarity. However, mammalian oocytes lack centrosomes. Our findings reveal an oocyte-specific kinetochore function as scaffolds to promote error-free acentrosomal spindle bipolarization in mice. Recruitment of the antiparallel microtubule crosslinker Prc1 to kinetochores mediated by the Ndc80 complex is a key molecular mechanism underlying this function. The female-meiosis-specific enrichment of Prc1 at kinetochore represents developmental plasticity of kinetochores. Prc1 has previously been characterized as an essential factor in the formation of the spindle midzone during anaphase in centrosomal cells[56–58]. In *Drosophila* oocytes, several microtubule regulators that work specifically during

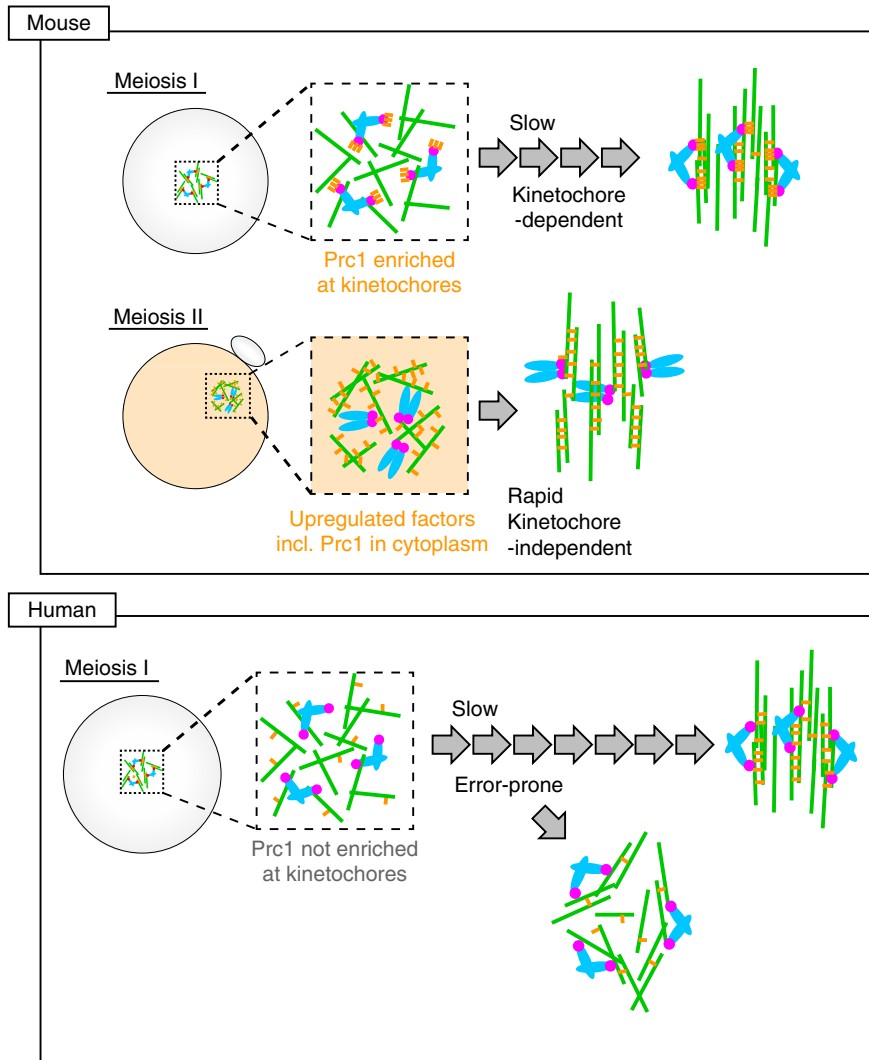

**Fig. 8 Models for acentrosomal spindle bipolarization in mammalian oocytes.** Different modes of spindle bipolarization. In mouse MI oocytes, Prc1 (orange) is highly enriched at kinetochores (magenta). The Prc1-rich kinetochore microenvironment is required for driving slow bipolarization of the spindle (green). This kinetochore-dependent spindle bipolarization is critical for MI to prevent chromosome segregation errors. In mouse MII oocytes, the cytoplasmic environment that contains upregulated factors including Prc1 provides global support to rapid, kinetochore-independent pathways for spindle bipolarization. In human MI oocytes, spindle bipolarization occurs without kinetochore-enriched Prc1, and is error prone.

anaphase in centrosomal cells localize at the central region of the bipolar spindle prior to anaphase I[64,65]. Thus, microtubule sorting in the central region of the spindle through anaphase-like mechanisms may be a conserved feature of acentrosomal spindle bipolarization[1,7]. Mouse oocytes, and not human oocytes, may have evolved strategies to enhance Prc1 enrichment at kinetochores for error-free spindle bipolarization.

## Methods

**Mice.** All animal experiments were approved by the Institutional Animal Care and Use Committee at RIKEN Kobe Branch (IACUC). Mice was housed in 12-h light/ 12-h dark cycle in temperature of 18–23 °C with 40–60% humidity environment. B6D2F1 (C57BL/6 × DBA/2), Ndc80[f/f], and Ndc80[f/f] Zp3-Cre (C57BL/6 back-ground) female mice, 8–12 weeks of age, were used to obtain oocytes. B6D2F1 males were used to obtain spermatocytes.

**Generation of Ndc80 conditional knockout mice.** We constructed a targeting vector that carries Ndc80 exon 2 from C57BL/6 genomic DNA flanked by loxP sites with an frt-flanked neomycin selection cassette (Supplementary Fig. 1a). The targeting vector was introduced into TT2 embryonic stem (ES) cells (C57BL/6 × CBA)[66]. After neomycin selection, the genomic DNAs of the clones were tested via Southern blot analysis with specific probes (5′: chr17:71538317-71538981, 3′: chr17:71509674-71510369) (Supplementary Fig. 1b). The ES cells that carried the targeted allele were

injected into ICR 8-cell-stage embryos and transferred to pseudopregnant mothers, which generated germline transmitters of the targeted allele. After at least 5 back-crosses to C57BL/6 mice, the resulting Ndc80[f] mice (Accession No. CDB1212K: http://www2.clst.riken.jp/arg/mutant%20mice%20list.html) were mated with Zp3-Cre mice[51] and the resulting Ndc80[f] Zp3-Cre were mated with Ndc80[f] mice to generate Ndc80[f/f] Zp3-Cre mice. We mated Ndc80[f/f] Zp3-Cre males with Ndc80[f/f] females to maintain the lines that produced Ndc80[f/f] Zp3-Cre females (used to obtain 'Ndc80-deleted' oocytes) and Ndc80[f/f] females (used to obtain control oocytes). Genotyping of Ndc80[f/f] was performed with the primers Ndc80-fw (GAGACC CCTTAGAACTTCTCCAGGCCAGGT), Ndc80-wt-rev (CAACGAGCTAAAACC TAGCATCGTGTCCAC), and Neo-rev (CGCCAAGTGCCCAGCGGGGCTGC TAAAGCG).

**Mouse oocyte culture.** Mice were injected with 5 IU of equine chorionic gona-dotropin (eCG, ASKA Pharmaceutical) or 0.1 ml of CARD HyperOva (KYUDO). Fully-grown oocytes at the germinal vesicle (GV) stage were collected 48 h after injection and released into M2 medium containing 200 nM 3-isobutyl-1-methyl-xanthine (IBMX, Sigma) at 37 °C. Meiotic resumption was induced by washing to remove IBMX. When indicated, 660 μM nocodazole (Sigma), 100 μM monastrol (Sigma), and 1 μM reversine (Cayman) were used.

**RNA microinjection.** mRNAs were transcribed in vitro using the mMESSAGE mMACHINE T7 kit (Thermo Fisher Scientific) and purified. The mRNAs were introduced into fully-grown GV-stage mouse oocytes through microinjection. The

microinjected oocytes were cultured at 37 °C for 3–4 h before meiotic resumption. Microinjections were performed with 3 pg of EGFP-Map4, 0.15 pg of H2B-mCherry, 2 pg of full-length Ndc80, 1 pg of Ndc80ΔN, 1 pg of Nuf2ΔN, 1 pg of Ndc80ΔN-4A-Spc25C, 1 pg of Nuf2ΔN-Spc24C, 0.15 pg of Securin-EGFP, 0.7 pg tdTomato-CENP-C, 4 pg of Ccnb2, 3 pg of mEGFP-HURP, 3 pg of mEGFP-Kif11, 2 pg of mNeonGreen-HSET, 0.1 pg of Prc1-mEGFP (Supplementary Fig. 6f, g), 2 pg of mNeonGreen-Prc1 (Prc1 overexpression for rescue, Fig. 5c), 2.5 pg of Prc1 (artificial increase in Prc1 to MII levels, Fig. 3), 0.1 pg of 24xGCN4-Prc1 together with 0.25 pg of scFv-sfGFP (SunTag-Prc1)[67], 0.1 pg of Ndc80-24xGCN4 together with 0.25 pg of scFv-sfGFP (Ndc80-SunTag), 4.5 pg of human Ndc80ΔN, 4.5 pg of human Nuf2ΔN, 1.3 pg of 24xGCN4-human Prc1 together with 4.5 pg of scFv-sfGFP (SunTag-human Prc1), 1 pg of Ndc80-sfGFP together with 1 pg of Nuf2 (Fig. 5a), 0.7 pg of mEGFP, and 2 pg of Ndc80-mEGFP (Fig. 4b).

**Beads injection.** Anti-GFP mAB-Magnetic Beads (D153-11, MBL) were introduced into mouse MI oocytes (1–2 h after NEBD) expressing mEGFP or Ndc80-mEGFP. Each oocyte was held by a holding pipette at the 9 o'clock position. The zona pellucida was punctured by several applications of piezo-pulse. Next, 7–10 beads, which had been aspirated into a glass micropipette, were ejected into the cytoplasm of the oocyte. The injection pipette was then slowly withdrawn. The MI recipient oocytes were cultured for 30 min to allow the binding of mEGFP-fused proteins.

**RNAi.** Short-interfering RNAs (siRNAs) were designed by siDirect (http://sidirect2.rnai.jp/). The sequences of the Prc1 siRNAs were siPrc1 (GGAAAUAUGGGAAC UAAUUGG and AAUUAGUUCCCAUAUUUCCCG) and siPrc1-2 (GAAAAA AUCACAAAAUGAAGC and UUCAUUUUUGUGAUUUUUUCAU). siPrc1 was used for the experiments shown in this paper. siPrc1-2 yielded phenotypes that were essentially identical to those of siPrc1 (data not shown). A luciferase siRNA (CGUACGCGGAAUACUUCGAUU and UUGCAUGCGCCUUAUGAAGCU) was used as a control. A total of 2pl of 760 μM siRNA was microinjected into fully-grown GV-stage oocytes. For live imaging and rescue experiments, mRNAs of fluorescent markers and rescue constructs were co-microinjected with the siRNAs. The quantity of mRNA used was 3 pg for EGFP-Map4, 0.15 pg for H2B-mCherry, and 1 pg for siRNA-resistant Prc1 (GGAAATATGGGAACTAATTGG mutated to AGAGAGATCTGGGAGTTGATC). The microinjected oocytes were cultured at 37 °C for 9 h before the induction of meiotic resumption.

**Oocyte fusion.** MI and MII oocytes were fused using an electrofusion method[68]. Zona-free oocytes at the GV stage and those in MII were treated with M2 medium containing 0.05 mg/ml phytohemagglutinin for 1 min and physically attached to each other using a micromanipulator. Tightly attached GV-MII oocytes were placed in a fusion chamber (with the electrodes 0.5 mm apart) filled with fusion buffer (0.3 M mannitol, 0.1 mM MgSO₄, and 0.1% polyvinyl-pyrrolidone). We used an LF101 Electro Cell Fusion Generator (NEPA GENE) with a 10-V alternate current at 1 MHz for 1 s to align the attached GV-MII oocytes and a 60-V direct current pulse for 20 μs to induce membrane fusion. Successful fusion was verified by observing NEBD of the GV, which was presumably induced by M-phase-promoting factors derived from the cytoplasm of the MII oocyte.

**Live imaging.** A customized Zeiss LSM710 or LSM880 confocal microscope equipped with a 40 × C-Apochromat 1.2NA water immersion objective lens (Carl Zeiss) was controlled by Zen software with the multiposition autofocus macros AutofocusScreen and MyPiC, which were developed by the group of Dr. Jan Ellenberg at EMBL Heidelberg (https://www.ellenberg.embl.de/resources/microscopyautomation). For imaging of spindles and chromosomes, we recorded 11 z-confocal sections (every 4 μm) of 512 × 512 pixel xy images, which covered a total volume of at least 53.1 μm × 53.1 μm × 40 μm, at 5- or 6-min time intervals for at least 12 h after the induction of maturation. For imaging of kinetochores, we recorded 17 z-confocal sections (every 1.5 μm) of 512 × 512 pixel xy images, which covered a total volume of at least 32.0 μm × 32.0 μm × 25.5 μm, at 5- or 6-min time intervals for at least 12 h after the induction of maturation. For imaging of fused oocytes, we recorded 12 z-confocal sections (every 6 μm) of 512 × 512 pixel xy images, which covered a total volume of at least 212.1 μm × 212.1 μm × 66 μm, at 5-min time intervals for at least 12 h after the fusion. Z-projection images are shown. We defined anaphase I onset based on the onset of abrupt spindle rearrangement (which was associated with centripetal movement of chromosomes and shrinkage of the microtubule mass) that was followed by cytokinesis and chromosome decondensation (Supplementary Movie 3). This timing was consistent with the anaphase timing defined based on Securin-GFP intensity changes (Supplementary Fig. 2c).

**4D spindle analysis.** To analyze spindle shapes, we performed 3D surface rendering of the signals of EGFP-Map4 or microtubules with Imaris software (Bitplane). The generated 3D surfaces were used to categorize spindle shapes based on the aspect ratio and surface irregularity. The aspect ratio was calculated by determining the ratio of the length to the width of an ellipsoid fitted to the generated 3D surface. The surface irregularity was calculated by determining the ratio of the volume of the fitted ellipsoid to that of the 3D surface. When the aspect ratio

was greater than 1.2 (MI) or 1.6 (MII) and the surface irregularity was smaller than 1.1, the spindle was categorized as a bipolar spindle. Otherwise, the spindle was categorized as an apolar/multipolar spindle. When a spindle maintained a bipolar state for 1 h or longer with no collapse prior to anaphase I onset or the end of imaging, the spindle was defined as a stable bipolar spindle.

**Immunostaining of oocytes.** Oocytes were fixed with 1.6% formaldehyde (methanol-free) in 10 mM PIPES (pH 7.0), 1 mM MgCl₂, and 0.1% Triton X-100 for 30 min. For HSET staining, we used 3.7% formaldehyde (methanol-free) for fixation. When indicated, oocytes were pretreated with cold M2 medium on ice for 10 min (Supplementary Figs. 5c and 10b) or for 1 min (Fig. 5b) before fixation. After fixation, the oocytes were washed and permeabilized with PBT (PBS supplemented with 0.1% Triton X-100) at 4 °C overnight. The oocytes were blocked with 3% bovine serum albumin (BSA)-PBT. After overnight incubation with primary antibodies at 4 °C, the oocytes were washed with 3% BSA-PBT and then incubated with secondary antibodies and 5 μg/ml Hoechst33342 for 2 h. The oocytes were washed again and stored in 0.01% BSA-PBS. The oocytes were imaged under a Zeiss LSM780 confocal microscope. In the experiments of Supplementary Figs. 5c and 10b, oocytes were incubated with secondary antibodies for overnight, and imaged under an LSM880 confocal microscope with AiryScan. To score aneuploidies, we counted the number of kinetochores in MII eggs. We recorded z-confocal sections (every 0.25–0.30 μm) of xy images (at a resolution of <0.10 μm per pixel) to capture all chromosomes of the entire oocyte using LSM780 or LSM880 Airyscan.

The following primary antibodies were used: rabbit anti-Ndc80 antiserum (1:2000, a gift from Dr. Robert Benezra), human anti-centromere antibodies (1:200, ACA, CS1058, Europa Bioproducts), a rabbit anti-Nuf2 antibody (1:500, ab230313, Abcam), a mouse anti-α-tubulin antibody (1:500, DM1A T6199, Sigma), a rat anti-GFP antibody (1:500, GF090R 04404-84, Nacalai), a rabbit anti-Prc1 (1:100, H-70 sc-8356, Santa Cruz), a mouse anti-Ndc80 (1:500, 9G3.23, GeneTex), a rabbit anti-Aurora A (pT288) (1:500, NB100-2371, NOVUS biologicals), a sheep anti-BubR1 (1:100, ab28192, Abcam), a mouse anti-pericentrin (1:500, 611814, BD Transduction Laboratories), a rabbit anti-HURP (1:50, sc-98809, Santa Cruz), a mouse anti-Mad2 (1:500, sc-65492, Santa Cruz), a rabbit anti-HSET (1:200, a gift from Dr. Renata Basto)[69], a mouse anti-GFP (1:500, ab1218, abcam), a rat anti-α-tubulin (1:2000, MCA77G, Bio-Rad), a rabbit anti-Kif11 (1:500, HPA010568, Sigma), and a rabbit anti-CENP-C (1:500, a gift from Dr. Yoshinori Watanabe)[70]. The secondary antibodies were Alexa Fluor 488 goat anti-mouse IgG (H + L) (A11029); goat anti-rabbit IgG (H + L) (A11034); goat anti-rat IgG (H + L) (A11006); Alexa Fluor 555 goat anti-rabbit IgG (H + L) (A21429); goat anti-human IgG (H + L) (A21433); Alexa Fluor 647 goat anti-human IgG (H + L) (A21445); goat anti-mouse IgG (H + L) (A21236); donkey anti-mouse IgG (H + L) (A31571) (1:500, Molecular Probes). Z-projection images are shown.

**Human oocyte experiments.** In Fig. 7a, we used oocytes (n = 16) donated specifically for research by volunteer donors (n = 3) following informed consent. This was approved by the Health Research Authority (UK), NRES Committee North East, Newcastle and North Tyneside1 Local Research Ethics Committee (REC reference 16/NE/0003). Vitrified oocytes were stored under a research license (R0152) from the UK Human and Fertilization Authority (HFEA). Oocytes were harvested at 26–30 h post hCG injection. Some were vitrified before fixation (n = 5) and the remainder were not (n = 11). In both cases, oocytes were fixed from 28 to 46 post hCG injection in order to obtain oocytes at MI and MII. Vitrification was performed using RapidVit Oocyte and RapidWarm Oocyte vitrification kits (Vitrolife).

In Fig. 7b and Supplementary Fig. 11c, we used immature oocytes that were not used for fertility treatment due to retardation in maturation after patients received a full explanation of the experiments and provided signed informed consent. The experiments were approved by institutional human research ethics committees at RIKEN (KOBE-IRB-13-22) and IVF Namba Clinic (2014-1), registered in Japan Society of Obstetrics and Gynecology (registry number 132), and carried out under these guidelines. Controlled ovarian stimulation was performed by using standard protocols with modifications according to the patients' medical history. Oocytes were picked up by transvaginal aspiration with an 18 gauge single lumen needle (Smiths Medical) 36 h after human chorionic gonadotropin (hCG) injection. All mature oocytes were inseminated by intracytoplasmic sperm injection for fertility treatment. Oocytes that were still immature at 42 h after hCG injection were donated for this study.

The immature oocytes were fixed 2–6 h after NEBD in 2% paraformaldehyde solution at 37 °C for 30 min. Immunostaining was performed by the same procedure that was used for mouse oocytes except for primary antibody incubation for 1–2 overnights. In Fig. 7a, oocytes were incubated in a mixture of Vectashield Mounting Medium with DAPI (Vector Laboratories) diluted with 0.01% BSA (1 μl + 4 μl 0.01% BSA) for 30 min before imaging.

**Immunostaining of cultured cells.** NIH3T3 cells were seeded on coverslips in DMEM supplemented with 10% BSA and incubated at 37 °C overnight. The cells were fixed with PBS containing 3.5% formaldehyde (methanol-free) at room temperature for 7 min, then permeabilized with KB buffer (10 mM Tris-HCl, pH

7.5, 150 mM NaCl, and 0.1% BSA) plus 0.2% Triton X-100 for 5 min, and washed three times with KB buffer for 2 min. Cells were incubated with primary antibodies in the KB buffer for 2 h at room temperature, washed three times, and incubated with secondary antibodies for 1 h at room temperature. The primary antibodies used were anti-Prc1 (1:100), ACA (1:100, 15-234, Antibodies Incorporated), and anti-α-tubulin (1:100). The secondary antibodies were Alexa Fluor 488 anti-rabbit, Alexa Fluor 555 anti-human, and Alexa Fluor 647 anti-mouse antibodies. DNA was counterstained with 20 μg/ml of Hoechst33342. Z-projection images are shown.

**Immunostaining of spermatocytes.** Seminiferous tubules collected from a male mouse were incubated with shaking in 6 ml of RPMI1640 medium containing 0.5 mg/ml collagenase at 33 °C for 20–30 min. The incubation was continued for another 20 min after the addition of 200 μl of 0.25% Trypsin-EDTA and 10 U of DNase I. Cells in the seminiferous tubules were dissociated with pipetting. Tissue aggregates were removed with a cell strainer (40-μm diameter). The dissociated cells were transferred to a 1.5-ml tube, and then washed with RPMI1640 medium containing 0.1 mg/ml soybean trypsin inhibitor (SBTI). The cells were collected by centrifugation at 90g for 5 min, and dissociated with 500 μl of PBS. A 100-μl drop of the dissociated cells were placed on a slide glass and stored at room temperature for 30 min. The supernatant of the drop was removed, and the cells were fixed with 4% paraformaldehyde in PBS at room temperature for 5 min. The cells were washed with PBS for three times, and then blocked with 3% BSA in PBS for 1 h. The cells were incubated with primary antibodies in 3% BSA in PBS for overnight, washed three times, and then incubated with secondary antibodies for overnight. The primary antibodies used were anti-Prc1 (1:500), anti-Scp3 (1:500, ab97672, Abcam), and ACA (1:500, 15-234, Antibodies Incorporated). The secondary antibodies were Alexa Fluor 488 anti-rabbit, Alexa Fluor 555 anti-human, and Alexa Flour 647 anti-mouse antibodies. DNA was counterstained with 20 μg/ml of Hoechst33342. Spermatocytes in MI were determined by Scp3 signals and meiotic configuration of chromosomes. Z-projection images are shown.

**Signal intensity quantification.** Fiji (https://fiji.sc/) was used to quantify fluorescent signals. To determine the Prc1 levels at kinetochores, the mean fluorescence intensity of Prc1 was measured around the peak of the signal and subtracted by the signal intensity at a cytoplasmic region near the kinetochore. Similarly, we measured the ACA signal of the same kinetochore. We calculated the ratio between the levels of Prc1 and ACA to determine the relative level of Prc1 at the kinetochores. For each oocyte, the intensities of at least 38 kinetochores were analyzed and averaged. If no centromeric enrichment was observed, signal intensities near chromosomes were measured.

**FRAP.** For FRAP experiments, we used oocytes expressing SunTag-Prc1 (24xGCN4-Prc1 coexpressed with scFv-sfGFP), Ndc80-SunTag (Ndc80-24xGCN4 coexpressed with scFv-sfGFP), or Ndc80-sfGFP and tdTomato-CENP-C 1.5 h after NEBD. Spot bleaching of a kinetochore was performed with a 488-nm Ar laser, followed by image acquisition. The fluorescent signals at the kinetochore were measured and background-subtracted. The recovery curve was normalized and fitted with an exponential curve to obtain the half-life of fluorescence recovery. Note that the recovery curves in this experiment (0–90 s) reported the turnover of Prc1 and Ndc80 at kinetochores rather than the turnover of scFv on 24xGCN4, because the binding of scFv to 24xGCN4 is stable ($t^{1/2}$ dissociation rate 5–10 min[67]). This was confirmed by data showing that the recovery curve of Ndc80-SunTag did not differ significantly from that of Ndc80-sfGFP (Fig. 5a).

**Western blotting.** Extracts were prepared by heating oocytes in sample buffer at 95 °C for 5 min and tested by Western blotting. The primary antibodies used were rabbit anti-Ndc80 (1:500), rabbit anti-Prc1 (1:200), goat anti-cyclin B2 (1:2000, AF6004, R&D Systems), and rabbit anti-actin (1:2000, ab1801, Abcam) antibodies. The secondary antibodies were horseradish peroxidase-conjugated anti-rabbit and anti-goat antibodies (1:2000).

**Yeast two-hybrid screening.** We designed a bait plasmid, namely, pBridge_BD-Ndc80ΔN_Nuf2ΔN, for a yeast two-hybrid screen. The plasmid carries a DNA fragment encoding mouse Ndc80ΔN (a.a. 461–642) fused to a DNA-binding domain under the GAL4 promoter, and a DNA fragment encoding the mouse Nuf2ΔN (a.a. 276–463) under the Met25 promoter, which was derived from the pBridge vector (630404, Clontech). The pBridge_BD-Ndc80ΔN_Nuf2ΔN plasmid, together with a cDNA library derived from mouse oocytes was transformed with the yeast Y2HGold Yeast Strain (Clontech, 630498). The transformed cells were cultured on selective SD agar plates (SD−Trp−Leu−His−Ade). Colonies were picked for plasmid collection and sequencing. The positive clones were confirmed by retransformation and growth test on selective (SD−Trp−Leu−His−Ade) and nonselective (SD−Trp−Leu) plates.

**Statistical analysis.** Graphs were generated, and statistical analyses were performed using Excel and GraphPad Prism. No statistical analysis was performed to predetermine the sample size. If not specified, significance was determined using a two-tailed, unpaired Student's t-test. Sample sizes and p-values are indicated in the figures and figure legends.

**Reporting summary.** Further information on research design is available in the Nature Research Reporting Summary linked to this article.

## Data availability
All data of this study are stored at the corresponding author and available on reasonable request. The source data for Figs. 1c, e, 2f, 3b–d, 4a–d, 5a–c, 6e–g and Supplementary Figs. 2a–e, 4a, 5b, c, 7b, d, 9c, d, 10a–d, 11b are available as a Source Data file.

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

## Acknowledgements

We thank Dr. K. Ishiguro for advice on spermatocyte immunostaining, Dr. T. Nakamura for advice on two-hybrid screening; R. Benezra for providing the Ndc80 antibody; R. Basto for HSET antibody; Y. Watanabe for CENP-C antibody; J. Ellenberg for providing a macro for automated microscopy; the imaging and genome analysis and animal facilities of RIKEN Kobe for technical support; H. Kiyonari and Y. Furuta for managing mouse genetic engineering facilities; Y. Watanabe and D. Sipp for reading the paper. We also thank our laboratory members, especially M. Kaido for assisting molecular experiments. S.N. was supported by the RIKEN JRA program. This work was supported by the research grants MEXT/JSPS KAKENHI JP16H06161/JP16H01226/JP18H05549 to T.S.K.; European Union Horizon 2020 program GermAge (grant number 634113) to M.H.; JSPS KAKENHI JP17K15069/JP19K06682 to S.Y.; JSPS KAKENHI 17K08144 to S.H.; and by RIKEN BDR and Center for Developmental Biology (CDB).

## Author contributions

S.Y. designed the study, performed almost all experiments using mouse oocytes, analyzed and interpreted the data, prepared the figures, and contributed to the paper writing. S.N. performed experiments using spermatocytes and cultured cells. L.L., M.C., and M.H. investigated oocytes of women with no history of infertility. S.H. and Y.N. performed other experiments using human oocytes. T.M. investigated human Ndc80–Prc1 interaction. A.C. assisted characterization of *Ndc80*-deleted oocytes. H.K. performed beads injection and assisted oocyte manipulations. T.A. and A.S. generated *Ndc80<sup>f</sup>* mice. T.S.K. designed, conceptualized, and supervised the project, interpreted the data, and wrote the paper with help from all authors.

## Competing interests

The authors declare no competing interests.
