## [Peer Review File · Nature Communications]

Reviewers' comments:

Reviewer #1 (Remarks to the Author):

Spindle morphogenesis in most oocytes occurs in the absence of centrosomes, contributing to the tendency of oocytes to segregate their chromosomes with errors. This is true in particular in human oocytes, causing aneuploidies potentially leading to congenital diseases. Deciphering this acentrosomal mode of spindle morphogenesis is thus of importance in terms of reproductive biology, but also in cell biology since some pathways favored by oocytes are used in pathological situations by somatic cells. In this manuscript, Yoshida and collaborators nicely and quite convincingly describe that kinetochores drive spindle bipolarization in meiosis I in mouse oocytes. They decipher some of the molecular mechanism at play (oocyte-specific enrichment of Prc1 at kinetochores via the Ndc80 complex) and importantly show that it is restricted to meiosis I (and not meiosis II). Not surprisingly (see below minor), this process is essential for preventing chromosome segregation errors at anaphase I since forcing meiosis I spindle morphogenesis into a meiosis II mode (rapid bipolarization) induces segregation defects. At last, they suggest that this pathway might be absent from human oocytes, explaining why spindle bipolarization is error-prone. The findings are novel and very interesting. Overall it is a very comprehensive well executed study using state of the art imaging techniques and quantitative imaging. The paper contains a large amount of high-quality data and represents a lot of work. However, I have some reservations about some weaker points of the manuscript.

Major:

The fact that the kinetochores drive spindle bipolarization in meiosis I is the main conclusion from the paper. Thus, it should be very strong. In my opinion, some data are missing to achieve that, especially eliminating other possibilities.

1. In the introduction line 60, the authors say that « while much is known about the structures that function as scaffolds to promote microtubule nucleation, very little is known about the structures that promote spindle bipolarization ». This is too strong and incomplete. Bipolarization also requires a balance between plus-end and minus-end motors, and of course microtubule nucleation. Letort Mol Biol Cell 2019 recently published a computational model of the early stages of acentriolar meiotic spindle assembly, showing predictions of diverse perturbations on spindle bipolarization and chromosome alignment and confronting them with published papers. These papers include: Bennabi EMBO Rep 2018 that shows the role of HSET for spindle bipolarization, Kolano PNAS 2012 for NuMA, Schuh Cell 2007/Clift Nat Commun 2015/Mailhes Mutat Res 2004/Breuer JCB 2010 for Eg 5, Breuer JCB 2010 for HURP, Baumann J Cell Sci 2017/Ma Mol Reprod Dev 2014/ for aMTOC, Bury JCB 2017 for Aurora. All these papers should be cited (and not only few of them). Importantly, some of these known bipolarization factors should be tested (localization, rescue) in the Ndc80 depleted oocytes to exclude their contribution to the Ndc80 KO phenotype. Indeed, in the Ndc80 KO, Prc1 is absent from the kinetochore but other proteins could be absent as well. One could imagine that in Ndc80 KO oocytes, some proteins important for MT nucleation and/or motor/map recruitment are absent/misplaced. The authors verified MT nucleation by measuring EGFP-Map4 intensity, but they need to show and quantify tubulin itself to exclude that MT density is altered since it is known to perturb spindle bipolarization. For map/motor recruitment, of course I am not asking them to look at all the known proteins required for spindle morphogenesis, but at least at some important ones for bipolarization (Eg5, HURP, HSET). In addition, the authors never look at aMTOCs in their Ndc80 depleted oocytes, they should, to exclude an aMTOC phenotype.

2. To really show that the Kinetochore (KT) is essential for spindle bipolarization, the authors should inject beads loaded with DNA and beads loaded with DNA and Prc1 in oocytes and compare spindle bipolarization (like in the Deng Dev Cell 2007 paper, but in MI).

3. I praise the authors for having done so much rescues. However, I would have used and analyze in priority rescues done with a fusion of the proteins of interest to the KT (like they did for some rescues,

see FigS7), to be able to separate a global effect in the cytoplasm from a local one at the KT.

- Could the authors rescue Ndc80 depleted oocytes with Nuf2-KTtethered and Ndc80-KTtethered, to exclude the effect of Ndc80/Nuf2 overexpression in the cytoplasm?
- Could they rescue Ndc80 depleted oocytes by addressing Prc1 to the KT specifically (and not only by overexpression in the cytoplasm)?

4. The experiment of fusion (Fig 5) is nice, but the caveat is that the oocyte doubles its volume, so all the components present only in MI or MII are diluted by two. Why not do the reverse, transform an MII oocyte into an MI oocyte, by rescuing Prc1 siRNA oocytes by Prc1 specifically bound to the KT, and see if spindle bipolarization resembles the one in MI?

5. The authors say that the kinetochore-dependent mode of spindle bipolarization is required for MI to prevent chromosome segregation errors in human oocytes. They do not show that it's required, just that Prc1 is not there. First, it could be there but not detected by Immunofluorescence. Second, since spindle morphogenesis is different in human and mouse, some other proteins could be missing/mislocalized in human. At last, to show that it is required, the authors should express in human oocytes a construct that localizes Prc1 at the KT and see if it rescues chromosome segregation errors (or at least forces human spindle morphogenesis to become mouse like in terms of bipolarization).

Minor:

- Why is anaphase advanced in Ndc80 depleted oocytes (occurring around 5h after NEBD, Fig1B)? does it reflect a problem of loading of SAC proteins on the KT, impeding the SAC to arrest these oocytes in meiosis I? Where are SAC proteins localized in these oocytes?
- What is the status of chromosome attachment to the spindle in Ndc80 depleted oocytes (basically Figure S4 with Ndc80 KO)? If there is no chromosome-MT attachment, are some chromosomes not attached to the spindle and lost in the cytoplasm? It looks like in Fig 1A and in some movies. What happens to these chromosomes? Do they segregate, since Separase is active?
- Also in Fig1A, how do the authors define/score Anaphase? In the movies, anaphase is not obvious, there is no visible chromosome separation (at least in the examples shown, maybe they should put a star or something in the frame corresponding to anaphase), and no separation of the triangular microtubule structure. Does cytokinesis occur? Could the authors show a movie of an Ndc80 KO oocyte in transmitted light?
- Could the authors comment more on the type of aneuploidies they observe in Ndc80 depleted oocytes, and maybe show some spreads corresponding to fig 1C?
- Where is Ndc80 in human oocytes not treated with colchicine?
- concerning lines 258 and 341, Bennabi EMBO Rep 2018 already showed that bipolarizing too early and skipping the ball stage is critical for chromosome alignment later on. Letort Mol Biol Cell 2019 examines exactly that. These papers should be cited and discussed.
- Line 327: the sentence should be modified, saying that these findings along others explain why spindle formation in MI takes longer than in MII. Indeed, HSET levels increase during MI (Bennabi EMBO Rep 2018), as well as Eg5 (Letort Mol Biol Cell 2019), TPX2 and P-TACC3 (Brunet Plos One 2008), MISS (Lefebvre JCB 2002) and maybe yet unknown factors...

Reviewer #2 (Remarks to the Author):

It is fast emerging that the machinery of chromosome segregation functions differently in mammalian oocyte meiosis-I. Many differences have been spotted, and are important as they may help explain the error-prone nature of this particular cell division. This paper reports that PRC1, best known as a MT crosslinker in anaphase in particular, may be found on the kinetochore in MI, as a result of a direct interaction with NDC80, and is necessary for spindle stability. This is definitely a departure from mitosis, and is thus interesting. My main hesitations are twofold. Firstly that the level of proof that the protein is kinetochore-bound needs to be higher (specific point 1 below). Secondly it is a shame that there is not a better explanation of why PRC1 is essential in MI (specific point 2 below).

1. The major finding that PRC1 is found on kinetochores in MI is unexpected and therefore requires further validation, particularly since it is then no longer seen on kinetochores in MII. Validation should include at least (a) immunofluorescence without a kinetochore counterstain, to rule out crosstalk/bleedthrough, and (b) demonstration using immunofluorescence that the signal disappears after PRC1 knockdown. In addition the following should be straightforward and would further support the point: (c) full length western, and (d) use of a traditional GFP tagged protein.
2. Assuming these simple controls validate the result – the question remains, why is it not at the kinetochore in MII? And why is it so essential in MI?
3. The notion that “prc1 is one of the cytoplasmic factors that supports NDC80-independent spindle function” on P10 would be best supported by fusing PRC1-knockdown oocytes with NDC80 knockouts.
4. How was aneuploidy scored, and what were the precise numbers of chromosomes? This must be stated.
5. How does PRC1 get from the kinetochore to the microtubules – and can the authors show that happening using FRAP or some other approach?

Minor points

1. The comment ‘shift the mode of spindle bipolarisation toward that of MII’ is overstatement, and so are the experiments presented in lines 252-258
2. Some of the imaging is not quite up to this lab’s usual standards, and the surface rendering does not seem to realistically reflect the images. See for example fig 4b
3. Some of the rhetoric and writing needs toning-down. It is an over-simplification to say that spindle assembly is kinetochore-led in oocytes, as it is known that enucleated oocytes make spindles. Similarly it is a little awkward that the authors cite the SunQY NDC80 paper but fail to mention they depleted NDC80.
4. Overall the discussion is too long and several of the interpretations are over the top. For example, these results do not (alone) explain why MI is long. Overall I’d ask the authors to make the discussion much more concise.

Reviewer #3 (Remarks to the Author):

This manuscript convincingly demonstrates that kinetochore recruitment of the microtubule crosslinker PRC1 by the NDC80 complex is crucial for bipolarity of the spindle in meiosis I in mouse oocytes. Furthermore, they provided interesting evidence suggesting that this kinetochore-dependent mode is important for accurate chromosome segregation, and the error-prone nature of human oocytes may be, at least partially, due to lack of PRC1 at kinetochores.

The spindle forms without centrosomes in oocytes in many animal species. How spindle bipolarity is established and maintained in oocytes is not yet understood. It is puzzling that establishment of spindle bipolarity commonly takes a long time in meiosis I in oocytes, and that spindle bipolarity is unstable in human meiosis I oocytes. The work presented in this manuscript addresses these

important issues. The conclusion is novel and unanticipated, and represents a significant advance in the field of oocyte meiosis. The manuscript is well written. It is easy to follow the rationale and the flow of the work is logical.

This is an excellent piece of work, which I would be very happy to see in, say, Nature Cell Biology. It may not have a flashy message, but contains very clear conclusions and further insights into important issues related to chromosome segregation and mis-segregation in mouse and human oocytes. What I really like is how the series of experiments is constructed. Logical development of the work is excellent and multi-layers of experiments were carried out to support the conclusion. Oocytes are known to be very difficult systems to study, but these experiments presented here go well beyond what is normally expected in oocytes in terms of both quality and quantity. I found no specific concerns in the manuscript, except a minor one detailed below. I strongly recommend publication in Nature Communications.

L256. Strictly speaking, these data only demonstrate that the restricted level of Prc1 is critical for slow bipolarization and error-free chromosome segregation. A causal relationship between slow bipolarization and error-free chromosome segregation is a reasonable, but unproven, hypothesis.

Point-by-point response to reviewer's comments

NCOMMS-19-27684-A

Prc1-rich kinetochores are required for error-free acentrosomal spindle bipolarization during meiosis I in mouse oocytes

*correspondence to: tomoya.kitajima@riken.jp

Italic letters indicate reviewers' comments.

Blue letters indicate our responses.

“Red letters enclosed in parentheses” indicate texts newly added to the revised manuscript.

Fig. xx in red letters indicate figures newly added or updated in the revised manuscript.

“Plain letters enclosed in parentheses” indicate texts quoted from the previous manuscript.

Abbreviations: KT, kinetochore; MI, meiosis I; MII, meiosis II; MTOC, microtubule organizing centers; SAC, spindle assembly checkpoint.

First of all, we thank all reviewers for their suggestions and comments, which greatly improved our manuscript.

Reviewer #1

Spindle morphogenesis in most oocytes occurs in the absence of centrosomes, contributing to the tendency of oocytes to segregate their chromosomes with errors. This is true in particular in human oocytes, causing aneuploidies potentially leading to congenital diseases. Deciphering this acentrosomal mode of spindle morphogenesis is thus of importance in terms of reproductive biology, but also in cell biology since some pathways favored by oocytes are used in pathological situations by somatic cells.

In this manuscript, Yoshida and collaborators nicely and quite convincingly describe that kinetochores drive spindle bipolarization in meiosis I in mouse oocytes. They decipher some of the molecular mechanism at play (oocyte-specific enrichment of Prc1 at kinetochores via the Ndc80 complex) and importantly show that it is restricted to meiosis I (and not meiosis II). Not surprisingly (see below minor), this process is essential for preventing chromosome segregation errors at anaphase I since forcing meiosis I spindle morphogenesis into a meiosis II mode (rapid bipolarization) induces segregation defects. At last, they suggest that this pathway might be absent from human oocytes, explaining why spindle bipolarization is error-prone. The findings

are novel and very interesting. Overall it is a very comprehensive well executed study using state of the art imaging techniques and quantitative imaging. The paper contains a large amount of high-quality data and represents a lot of work.

However, I have some reservations about some weaker points of the manuscript.

We thank the reviewer for the appreciation of the quality and novelty of our work, and for the constructive comments.

Major:

The fact that the kinetochores drive spindle bipolarization in meiosis I is the main conclusion from the paper. Thus, it should be very strong. In my opinion, some data are missing to achieve that, especially eliminating other possibilities.

1. In the introduction line 60, the authors say that « while much is known about the structures that function as scaffolds to promote microtubule nucleation, very little is known about the structures that promote spindle bipolarization ». This is too strong and incomplete. Bipolarization also requires a balance between plus-end and minus-end motors, and of course microtubule nucleation. Letort Mol Biol Cell 2019 recently published a computational model of the early stages of acentriolar meiotic spindle assembly, showing predictions of diverse perturbations on spindle bipolarization and chromosome alignment and confronting them with published papers. These papers include: Bennabi EMBO Rep 2018 that shows the role of HSET for spindle bipolarization, Kolano PNAS 2012 for NuMA, Schuh Cell 2007/Clift Nat Commun 2015/Mailhes Mutat Res 2004/Breuer JCB 2010 for Eg 5, Breuer JCB 2010 for HURP, Baumann J Cell Sci 2017/Ma Mol Reprod Dev 2014/ for aMTOC, Bury JCB 2017 for Aurora. All these papers should be cited (and not only few of them). Importantly, some of these known bipolarization factors should be tested (localization, rescue) in the Ndc80 depleted oocytes to exclude their contribution to the Ndc80 KO phenotype. Indeed, in the Ndc80 KO, Prc1 is absent from the kinetochore but other proteins could be absent as well. One could imagine that in Ndc80 KO oocytes, some proteins important for MT nucleation and/or motor/map recruitment are absent/misplaced. The authors verified MT nucleation by measuring EGFP-Map4 intensity, but they need to show and quantify tubulin itself to exclude that MT density is altered since it is known to perturb spindle bipolarization. For map/motor recruitment, of course I am not asking them to look at all the known proteins required for spindle morphogenesis, but at least at some important ones for bipolarization (Eg5, HURP, HSET). In addition, the authors never look at aMTOCs in their Ndc80 depleted oocytes, they should, to exclude an aMTOC phenotype.

We thank the reviewer for these helpful comments and for suggesting important experiments. In the revised manuscript, we have addressed all the concerns as follows.

First, we have corrected our too strong sentences in Introduction. The revised manuscript now states “While much is known about the structures that function as scaffolds to promote microtubule nucleation, **less** is known about the structures that promote spindle bipolarization.” (line 61–63). Accordingly, we have revised a similar sentence in Summary (line 27–29).

Second, the revised manuscript comprehensively introduces the current knowledge of microtubule regulators that contribute to spindle bipolarization by stating “many acentriolar microtubule organizing centers (**MTOCs**), which are activated around chromosomes through **well-studied pathways including** the RanGTP pathway, act as major sites for microtubule nucleation (Dumont et al, J Cell Biol, 2007; Schuh and Ellenberg, Cell, 2007; Breuer et al, J Cell Biol, 2010; **Ma and Viveiros, Mol Reprod Dev, 2014; Clift and Schuh, Nat Commun, 2015; Baumann et al, J Cell Sci 2017; Bury et al, J Cell Biol, 2017)**.....**The balanced activities of microtubule regulators, such as the plus-end-directed motor Kif11 (also known as kinesin-5 and Eg5), the minus-end-directed motor HSET (kinesin-14), the bundle stabilizer HURP, and the minus-end clustering factor NuMA, that act on microtubules are critical for spindle bipolarization (Mailhes et al, 2004, Mutat Res Genetic Toxicol Environ Mutagen; Schuh and Ellenberg, Cell, 2007; Breuer et al, J Cell Biol, 2010; Kolano et al, PNAS, 2012; Clift and Schuh, Nat Commun, 2015; Bennabi et al, EMBO Rep, 2018; Letort et al, Mol Biol Cell, 2019). However, scaffolds that primarily enrich spindle bipolarization factors independently of microtubules have not been identified.**” (line 59–68), which includes citations to all the suggested papers.

Third, the revised manuscript now shows the distribution of aMTOCs. In control oocytes, pericentrin-marked aMTOCs distributed around the microtubule mass at prometaphase I, and then relocated to the poles of the forming bipolar spindle at metaphase I, as previously reported (Schuh and Ellenberg, Cell 2007; Breuer et al, J Cell Biol 2010; Clift and Schuh, Nat Commun 2015). In *Ndc80*-deleted oocytes, at prometaphase I, the total volume of aMTOCs were comparable to those in control oocytes (**Fig. S2b and S8**). At metaphase I, aMTOCs remained around the microtubule mass, associated with spindle bipolarization defects of *Ndc80*-deleted oocytes. At metaphase II, aMTOCs normally localized at spindle poles in *Ndc80*-deleted oocytes (**Fig. S8**). Moreover, *Ndc80* deletion did not significantly affect the localization of Aurora A at aMTOCs (**Fig. S8**). These results suggest that *Ndc80* is not directly involved in the formation of functional aMTOCs.

We describe these results in the revised manuscript “the total volume of aMTOCs and the kinetics of microtubule nucleation upon NEBD were largely intact (Fig. 1a, S2a and S2b)” (line 135–136), “the localizations of...Aurora A, microtubule regulators that contribute to spindle bipolarization (...Bury et al, J Cell Biol, 2017...), were not directly affected by *Ndc80* deletion (Fig. S8)” (line 219–221).

Figure S2b

Figure S2b: MTOC volume is largely intact in *Ndc80*-deleted oocytes. Oocytes 1 h after NEBD were immunostained for microtubules (anti-tubulin, green), MTOCs (anti-pericentrin, Pcmt, magenta) and counterstained for DNA (Hoechst33342, blue). Measurements were performed after 3D reconstruction (n = 18, 18 oocytes from 3 independent experiments). n.s., not significant by Student's t-test. Scale bar, 10 μ m. Error bars show the SD.

Figure S8

Figure S8: *Ndc80* deletion does not directly affect the localization phosphorylated Aurora A or pericentrin (Pcmt). Oocytes 2 h (Prometaphase I), 5 h (Metaphase I), 16 h (Metaphase II) after NEBD were immunostained for phosphorylated Aurora A (pAurora A), or pericentrin (Pcmt), tubulin (microtubules, green), and DNA (Hoechst33342, blue). Scale bars, 10 μ m.

Forth, we tested the localizations of HSET, Kif11 (Eg5), HURP in *Ndc80*-deleted oocytes. In control oocytes, HSET and HURP localized at spindle microtubules, and Kif11 localized at spindle poles, as previously reported (Breuer et al, J Cell Biol 2010, Clift and Schuh, Nat Commun 2015; Bennabi et al, EMBO Rep 2018). These localization patterns were not affected in *Ndc80*-deleted oocytes (Fig. S8). These results indicate that *Ndc80* is not directly involved in the localization of these proteins.

We describe these results in the revised manuscript “the localizations of HSET, Kif11, HURP..., microtubule regulators that contribute to spindle bipolarization (Breuer et al, 2010, J Cell Biol; Clift and Schuh, 2015; Bennabi et al, EMBO Rep, 2018...), were not directly affected by *Ndc80* deletion (Fig. S8)” (line 219–221).

Figure S8

Figure S8: *Ndc80* deletion does not directly affect the localization of HSET, Kif11 or HURP. Oocytes 2 h (Prometaphase I), 5 h (Metaphase I), 16 h (Metaphase II) after NEBD were immunostained for HSET, Kif11, HURP, tubulin (microtubules, green), and DNA (Hoechst33342, blue). Scale bars, 10 μ m.

Fifth, we tested whether overexpression of HSET, Kif11, or HURP rescues spindle bipolarization defects in *Ndc80*-deleted oocytes. While *Prc1* overexpression efficiently rescued spindle bipolarization defects in *Ndc80*-deleted oocytes, neither of the overexpression of HSET or Kif11 significantly rescued the defects (Fig. 5c). Overexpression of HURP slightly, although not significantly, rescued the defects (Fig. 5c), which may suggest functional overlap between HURP and *Prc1*, both of which act in microtubule bundling.

We describe these results in the revised manuscript “neither of the overexpression of HSET or Kif11 significantly rescued the spindle defects in *Ndc80*-deleted oocytes (Fig. 5c). HURP

overexpression slightly, although not significantly, rescued the spindle defects (Fig. 5c), which may suggest functional overlap between HURP and Prc1⁴ (line 244–247).

Figure 5c

Figure 5c: Prc1 overexpression rescues spindle defects in *Ndc80*-deleted oocytes. *Ndc80^{ff} Zp3-Cre* oocytes overexpressing mEGFP-HURP, mEGFP-Kif11, mNeonGreen-HSET, or mNeonGreenmNeonGreen-Prc1 were immunostained at metaphase I (5.5 h after NEBD). Spindle shapes were reconstructed in 3D and categorized (n = 29, 18, 18, 18, 24 oocytes from 3 independent experiments). ****p<0.0001 by Student's t-test. n.s., not significant. Scale bar, 10 μ m. Error bars show the SD.

Lastly, we quantified microtubule density by immunostaining tubulin itself in *Ndc80*-deleted oocytes. The microtubule density at prometaphase I in *Ndc80*-deleted oocytes was comparable to that in control oocytes (Fig. S2b).

Figure S2b

Figure S2b: Microtubule density is largely intact in *Ndc80*-deleted oocytes. Oocytes 1 h after NEBD were immunostained for microtubules (anti-tubulin, green), MTOCs (anti-pericentrin, Pcmt, magenta) and counterstained for DNA (Hoechst33342, blue). Measurements were performed after 3D reconstruction (n = 18, 18 oocytes from 3 independent experiments). n.s., not significant by Student's t-test. Error bars show the SD.

2. To really show that the Kinetochore (KT) is essential for spindle bipolarization, the authors should inject beads loaded with DNA and beads loaded with DNA and Prc1 in oocytes and compare spindle bipolarization (like in the Deng Dev Cell 2007 paper, but in MI).

We thank the reviewer for suggesting this experiment. This experiment would be a test for whether tethered Prc1 is sufficient for spindle bipolarization. First, we tested whether DNA beads (with no Prc1) are sufficient to induce microtubule nucleation in MI, as reported in MII (Deng et al, Dev Cell, 2007). While we could confirm that DNA beads sufficiently induced microtubule nucleation when microinjected into MII oocytes, the DNA beads nucleated no or few microtubules when microinjected into MI oocytes (Figure for reviewer 1). These results suggest that, unlike during MII, microtubule nucleation during MI requires not only DNA-mediated pathway but also additional factors. The requirement of additional factors for efficient microtubule nucleation in MI may reflect the importance of RanGTP-independent pathways (Dumont et al, J Cell Biol 2007; Bury et al, J Cell Biol 2017). Further supporting this idea, microtubule nucleation was not observed even when metaphase II chromosomes were transplanted to MI oocytes, whereas efficient microtubule nucleation and spindle formation were observed when chromosomes were placed back to MII oocytes (Figure for reviewer 2). The fact that DNA beads (or even intact chromatin) are insufficient to induce microtubule nucleation in MI precluded testing the effects of tethered Prc1 on spindle bipolarization in this experimental system.

Figure for reviewer 1

Microtubules nucleation on DNA beads

n=47 (MI), 46 (MII) oocytes

Oocytes were fixed at 4h after injection of DNA beads

Figure for reviewer 1: DNA beads efficiently induce microtubule nucleation in MII oocytes but not in MI oocytes. DNA beads were introduced into MI or MII oocytes. Oocytes were incubated for 4 hours, and then fixed for immunostaining of microtubules. While we observed efficient microtubule nucleation and spindle formation around DNA beads in MII oocytes, we did not observe efficient microtubule nucleation in MI oocytes. Scale bar, 10 μ m.

Figure for reviewer 2

Figure for reviewer 2: Chromatin efficiently induces microtubule nucleation in MII oocytes but not in MI oocytes. MII chromatin was collected using micropipettes and introduced to MI (GV stage) or MII (metaphase II stage) oocytes through electrofusion. Oocytes were incubated for >1 hours. Introduced chromatin failed to nucleate microtubules (arrowheads) in MI oocytes, whereas introduced chromatin nucleated microtubules and formed a spindle in MII oocytes. Scale bars, 10 μ m.

As the reviewer may have concerned, we currently lack strong evidence that concentrated Prc1 is sufficient for driving spindle bipolarization during MI, except that tethering Prc1-enriching domains of Ndc80/Nuf2 rescues spindle bipolarization in *Ndc80*-deleted oocytes in a Prc1-dependent manner (Fig. 4). We therefore do not exclude the possibility that Prc1 requires other factors at kinetochores to drive spindle bipolarization in MI. In the revised manuscript, we have corrected several statements that may have read as if the concentrated Prc1 is sufficient for driving spindle bipolarization during MI. In particular, we now avoid the use of our previous statement “Prc1 drives spindle bipolarization”, which was too strong, as the reviewer concerns. Accordingly, we revised the title of the manuscript, which is now “Prc1-rich kinetochores **are required for** error-free acentrosomal spindle bipolarization during meiosis I in mouse oocytes”. This main conclusion is robustly supported by our results including (1) the deletion of the Prc1 kinetochore recruiter Ndc80 causes spindle bipolarization defects, (2) tethering Prc1-enriching domains at kinetochores promotes spindle bipolarization in *Ndc80*-deleted oocytes in a Prc1-dependent manner, and (3) Prc1 knockdown delays spindle bipolarization. We also clearly state

that “kinetochores act as a scaffold to increase the local concentration of Prc1 and possibly other microtubule regulators that cooperatively promote spindle bipolarization.” (line 326–328).

3. *I praise the authors for having done so much rescues. However, I would have used and analyze in priority rescues done with a fusion of the proteins of interest to the KT (like they did for some rescues, see FigS7), to be able to separate a global effect in the cytoplasm from a local one at the KT.*

- Could the authors rescue Ndc80 depleted oocytes with Nuf2-KTtethered and Ndc80-KTtethered, to exclude the effect of Ndc80/Nuf2 overexpression in the cytoplasm?

We thank the reviewer for raising this important point. To separate a global effect in the cytoplasm from a local one at kinetochores, we need a two-step strategy to design a Nuf2- or Ndc80-KT-tethered construct: (1) identify a Nuf2 or Ndc80 fragment that has a Prc1-interacting capacity but no kinetochore-localizing capacity, (2) fuse an artificial kinetochore-tethering domain to the fragment. We found the step 1 challenging. Our comprehensive set of rescue experiments using various Ndc80/Nuf2 fragments and mutants indicated that the two capacities of Ndc80/Nuf2, kinetochore localization and Prc1 interaction, are tightly coupled at the C-terminal domain and therefore difficult to be separated. Nevertheless, we successfully identified one mutant fragment, Ndc80-4A, which retained Prc1 interaction but lost kinetochore localization (Fig. S9a and S9b). This fragment did not rescue spindle defects in *Ndc80*-deleted oocytes, but did rescue when it was tethered to kinetochores by fusion to Spc24/25C (Fig. 4c). We believe that these are best experiments that address the reviewer’s concern, and the results strongly support our conclusion that Ndc80/Nuf2-Prc1 locally acts at kinetochores, but not globally at the cytoplasm, for spindle bipolarization. The previous manuscript showed these data as a Supplementary Figure, but the revised manuscript now shows these in the main figure (Fig. 4c).

Figure 4c

Figure 4c: Ndc80/Nuf2–Prc1 acts at kinetochores. *Ndc80^{ff} Zp3-Cre* oocytes expressing Ndc80ΔN-4A/Nuf2ΔN (Y564A, Q565A, L566A, and T567A mutations; indicated as ‘NNΔN-4A’) or its Spc-fused form Ndc80ΔN-4A-Spc25C/Nuf2ΔN-Spc24C (indicated as ‘NNΔN-4A tethered at KT’) were monitored. Spc25C (a.a. 120–226) and Spc24C (a.a. 122–201) are kinetochore-targeting domains. Images of EGFP-Map4 (microtubules, green) and H2B-mCherry (chromosomes, magenta) at metaphase I (5.5 h after NEBD) are shown. Microtubule signals were reconstructed in 3D to visualize spindle shapes, which were categorized based on the aspect ratio and surface irregularity. The frequency of oocytes that exhibited a bipolar-shaped spindle is shown (n = 19, 18 oocytes from 3 independent experiments). ***p=0.0004 by Student’s t-test. Scale bar, 10 μm. Error bars show the SD.

- Could they rescue *Ndc80* depleted oocytes by addressing *Prc1* to the KT specifically (and not only by overexpression in the cytoplasm)?

We thank the reviewer for asking this experiment. This experiment would be a test for whether kinetochore-tethered Prc1 is sufficient for driving spindle bipolarization in *Ndc80*-deleted oocytes. We tethered Prc1 at kinetochores by fusing to Nsl1, a constitutive kinetochore component. As expected, the Prc1-Nsl1 fusion localized to kinetochores in oocytes (Figure for reviewer 3). We then evaluated the capacity of this kinetochore-Nsl1-tethered Prc1 to rescue spindle defects in *Ndc80*-deleted oocytes. However, we found that Nsl1-mediated tethering did not significantly increase the capacity of Prc1 to rescue (Figure for reviewer 3).

There are two, not mutually exclusive, explanations for why kinetochore tethering failed to enhance the capacity of Prc1 to rescue spindle bipolarization in *Ndc80*-deleted oocytes. First, the dynamic turnover of Prc1 at kinetochores (Fig. 5a) may be required to drive spindle bipolarization. This idea is consistent with our conclusion that “Ndc80/Nuf2 complex promotes spindle bipolarization by concentrating a dynamic pool of Prc1 at kinetochores” (line 248–249). Second,

Prc1 may require other Ndc80-dependent kinetochore factors to drive spindle bipolarization. We do not exclude the latter hypothesis, since there is no strong evidence that concentrated Prc1 is sufficient to drive spindle bipolarization. We therefore have revised our manuscript, in which we conclude that “Prc1-rich kinetochores **are required** for error-free spindle bipolarization” (revised title). Accordingly, we have revised our statements that may have read as if the concentrated Prc1 is sufficient for driving spindle bipolarization during MI, as described in our response to the comment 2.

Figure for reviewer 3

Figure for reviewer 3: Kinetochores tethering does not enhance the capacity of Prc1 to rescue spindle defects in *Ndc80*-deleted oocytes. Prc1 fused to the kinetochore component Nsl1 (Prc1-mNeonGreen-Nsl1) localizes to kinetochores in *Ndc80*-deleted oocytes (*Ndc80^{fl/fl} Zp3-Cre*). Bipolar spindle formation was scored by

immunostaining of microtubules at metaphase I. In an experimental condition where Prc1-mNeonGreen rescued spindle defects in *Ndc80-deleted* oocytes, Prc1-mNeonGreen-Nsl1 failed to rescue. Scale bars, 10 μ m.

4. The experiment of fusion (Fig 5) is nice, but the caveat is that the oocyte doubles its volume, so all the components present only in MI or MII are diluted by two. Why not do the reverse, transform an MII oocyte into an MI oocyte, by rescuing Prc1 siRNA oocytes by Prc1 specifically bound to the KT, and see if spindle bipolarization resembles the one in MI?

We thank the reviewer for suggesting this experiment. The possibility that kinetochore-tethered Prc1 in Prc1-depleted oocytes transforms MII spindle bipolarization into MI-like would be interesting, but the fact that kinetochore tethering of Prc1 did not rescue *Ndc80-deleted* oocytes (Figure for reviewer 3) precluded this possibility.

In our opinion, based on very important previous findings that MII cytoplasm contains a number of upregulated spindle factors (e.g. HSET, Eg5, TPX2, p-TACC3, MISS) (Lefebvre et al, J Cell Biol, 2002; Bennabi et al, EMBO Rep, 2018; Brunet et al, Plos One, 2018; Letort et al, Mol Biol Cell, 2019) and that HSET overexpression accelerates spindle bipolarization in MI (Bennabi et al, EMBO Rep, 2018; Letort et al, Mol Biol Cell, 2019) as well as Prc1 overexpression, it is likely that transforming spindle bipolarization from the MII-mode into the MI-mode would require the depletion of not only Prc1 but also other MII-upregulated proteins. Our idea is that the repertoire of proteins in the MII cytoplasm supports the MII-mode of spindle bipolarization, and that Prc1 is one of the members in the repertoire, as we discussed “Factors that contribute to the cytoplasmic support of MII include upregulated Prc1, and possibly other upregulated microtubule regulators, such as Tpx2 and Miss” in the previous manuscript.

In the revised manuscript, we now introduce the above previous findings “Several microtubule regulators such as HSET, Kif11, and Tpx2, Miss, and the phosphorylation of Tacc3 are upregulated in MII (Lefebvre et al, J Cell Biol 2002; Bennabi et al, EMBO Rep, 2018; Brunet et al, Plos One, 2018; Letort et al, Mol Biol Cell, 2019), which may contribute to rapid spindle bipolarization. Consistent with this idea, HSET overexpression accelerates spindle bipolarization in MI (Bennabi et al, EMBO Rep, 2018; Letort et al, Mol Biol Cell, 2019)” in Introduction (line 90–93). Moreover, we have revised several statements that might have read as if MII-upregulated Prc1 is only the critical factor that supports the MII-mode of spindle bipolarization. In particular, we discussed “...MII spindles form rapidly and are largely independent of functional kinetochores, with support from the cytoplasm that contain upregulated spindle bipolarizers including Prc1. These findings, along with previous findings that several microtubule regulators are upregulated in MII (Lefebvre et al, J Cell Biol 2002; Bennabi et al, EMBO Rep, 2018; Brunet et al, Plos One, 2018; Letort et al, Mol Biol Cell, 2019), at least partly explain why spindle formation in MI takes longer than that in MII” (line 335–339).

5. The authors say that the kinetochore-dependent mode of spindle bipolarization is required for MI to prevent chromosome segregation errors in human oocytes. They do not show that it's required, just that Prc1 is not there. First, it could be there but not detected by Immunofluorescence. Second, since spindle morphogenesis is different in human and mouse, some other proteins could be missing/mislocalized in human. At last, to show that it is required, the authors should express in human oocytes a construct that localizes Prc1 at the KT and see if it rescues chromosome segregation errors (or at least forces human spindle morphogenesis to become mouse like in terms of bipolarization).

We thank the reviewer for raising this point. To clarify, as the reviewer correctly points out, we did not experimentally show that the kinetochore-dependent mode of spindle bipolarization is required for preventing errors in human oocytes. In the previous manuscript, we therefore never stated that it is required in human. Instead, we discussed the possibility that the absence of detectable kinetochore Prc1 “may give a molecular explanation to why human but not mouse oocytes undergo error-prone spindle bipolarization” (line 368–369). Testing this hypothesis by manipulating the mode of spindle bipolarization in human oocytes will be a target of future studies. We fully agree with the reviewer that an undetectable pool of Prc1 may be located at kinetochores in human oocytes. We also agree that some other factors are missing in human. Indeed, the previous manuscript showed that human Ndc80 can recruit human Prc1 to kinetochores when expressed in *Ndc80*-deleted mouse oocytes (Fig. S11b), suggesting that unknown mouse-specific factors enhance Ndc80-dependent enrichment of Prc1 at kinetochores.

To further clarify these points in the manuscript, we have revised the text of the manuscript. First, we have added a clear statement that “our data do not exclude the possibility that an undetectable pool of Prc1 at kinetochores contribute to spindle bipolarization (in human oocytes)” (line 367–368). Second, “Future studies are needed to explore mouse-specific factors that enhance Prc1 enrichment at kinetochores” (line 372–374).

Minor:

- Why is anaphase is advanced in *Ndc80* depleted oocytes (occurring around 5h after NEBD, Fig1B)? does it reflect a problem of loading of SAC proteins on the KT, impeding the SAC to arrest these oocytes in meiosis I? Where are SAC proteins localized in these oocytes?

Thank you for pointing out this. *Ndc80*-deleted oocytes have no active SAC, which is consistent with *Ndc80* knockdown phenotypes in somatic cells (McClelland et al, Gene Dev 2003; Meraldi et

al, Dev Cell, 2004) and in oocytes (Sun et al, Cell Cycle, 2011). The revised manuscript now includes our data indicating that the SAC components, Mad2 and BubR1, are lost at kinetochores in *Ndc80*-deleted oocytes (Fig. S2d), and the statement that “The *Ndc80*-deleted oocytes then underwent aberrant anaphase I...consistent with spindle checkpoint defects (Fig. S2d) (McClelland et al, Gene Dev 2003; Meraldi et al, Dev Cell, 2004; Sun et al, Cell Cycle, 2011)” (line 140–142).

Figure S2d

Figure S2d: Spindle checkpoint defects in *Ndc80*-deleted oocytes. Oocytes 1 h after NEBD were immunostained for BubR1 (green), active form Mad2 (C-Mad2, green), CENP-C (magenta), and counterstained for DNA (Hoechst33342, blue). In the plot, spots correspond to individual kinetochores (n = 200, 194 kinetochores from 5, 5 oocytes). ****p < 0.0001 by Student's t-test. Scale bar, 10 μ m. Error bars, SD.

- What is the status of chromosome attachment to the spindle in *Ndc80* depleted oocytes (basically Figure S4 with *Ndc80* KO)? If there is no chromosome-MT attachment, are some chromosomes not attached to the spindle and lost in the cytoplasm? It looks like in Fig 1A and in some movies. What happens to these chromosomes? Do they segregate, since Separase is active?

Ndc80-deleted oocytes show almost no stable chromosome-microtubule attachment (please see the figure now in Fig. S5c, '*Ndc80*^{fl/fl} Zp3-Cre + *Ndc80* Δ N/*Nuf2* Δ N', which has no microtubule-binding domains and indeed shows almost no kinetochore-microtubule attachments). As you point out, in *Ndc80*-deleted oocytes, some chromosomes fell off the microtubule mass during MI, and underwent aberrant anaphase I. The revised figure now highlights these chromosomes with asterisks (Fig. 1a). To address the consequence of these chromosomes, we analyzed high-resolution confocal images covering the entire volume of *Ndc80*-deleted oocytes at MII. We found that chromosomes that were lost in the cytoplasm preferentially exhibited four kinetochores (Fig. S2e and S3). These data suggest that these chromosomes underwent nondisjunction at

anaphase I, likely due to the lack of microtubule-mediated bipolar pulling forces. We explicitly state the movement and consequence of these chromosomes, "Consistent with severe kinetochore–microtubule attachment defects, some chromosomes fell off the microtubule mass (Fig. 1a).....resulted in egg aneuploidies, including those derived from chromosome nondisjunction (Fig. 1c, S2e and S3)" (line 138–144).

Figure 1a

Figure 1a: Live imaging of *Ndc80^{fl/fl}* (control) and *Ndc80^{fl/fl} Zp3-Cre* (*Ndc80*-deleted) oocytes expressing EGFP-Map4 (microtubules, green) and H2B-mCherry (chromosomes, magenta). Spindle shapes were reconstructed in 3D. Asterisks show chromosomes that fell off the spindle. Scale bar, 10 μm.

Figure S2e

Figure S2e: Aneuploidy in *Ndc80*-deleted oocytes. Oocytes at metaphase II were immunostained for kinetochores (ACA, green) and counterstained for DNA (Hoechst33342, magenta). Arrowheads indicate four kinetochores (magnified), presumably derived from chromosome nondisjunction at anaphase I. Scale bar, 10 μm.

Figure S3

Figure S3: Kinetochore counting in MII eggs. Oocytes at metaphase II were immunostained for kinetochores (ACA, green) and counterstained for DNA (Hoechst33342, magenta). Optical slice images in a z-stack of *Ndc80^{fl}/Zp3-Cre* oocytes are shown. Numbers indicate kinetochore counts. Scale bar, 10 μ m.

- Also in Fig1A, how do the authors define/score Anaphase? In the movies, anaphase is not obvious, there is no visible chromosome separation (at least in the examples shown, maybe they should put a star or something in the frame corresponding to anaphase), and no separation of the triangular microtubule structure. Does cytokinesis occur? Could the authors show a movie of an *Ndc80* KO oocyte in transmitted light?

Thank you for these questions. We defined anaphase I onset based on the onset of abrupt spindle rearrangement (which was associated with centripetal movement of chromosomes and shrinkage of the microtubule mass, consistent with what would be expected if monopolar-like spindles underwent anaphase) that was followed by cytokinesis and chromosome decondensation. This timing was consistent with the anaphase timing defined based on Securin intensity changes (Fig. S2c).

In the revised manuscript, we now clearly state this definition in the legend of Fig. 1a and Materials and Methods (line 1180–1185). We also show a movie of an *Ndc80*-deleted oocyte, with a channel for transmitted light, in which abrupt spindle rearrangement followed by cytokinesis and chromosome condensation are obvious (Video S3).

- Could the authors comment more on the type of aneuploidies they observe in *Ndc80* depleted oocytes, and maybe show some spreads corresponding to fig 1C?

Thank you for these comments. We now present a plot showing the precise numbers of kinetochores in *Ndc80*-deleted MII eggs, which indicated various abnormalities including both loss and gain of chromosomes (Fig. S2e). As we described in the above response, we found that chromosomes that were lost in the MII cytoplasm preferentially showed four kinetochores, suggesting that these were derived from chromosome nondisjunction (Fig. S2e and S3). In these experiments, we used high-resolution confocal imaging of the entire volume of whole-mount oocytes, which allowed us to precisely and robustly detect 40 kinetochores in control oocytes (Fig. S2e).

Figure S2e

Figure S2e: Aneuploidy in Ndc80-deleted oocytes. Oocytes at metaphase II were immunostained for kinetochores (ACA, green) and counterstained for DNA (Hoechst33342, magenta). The number of kinetochores were counted (n = 28, 28 oocytes). Error bars, SD.

- Where is Ndc80 in human oocytes not treated with colchicine?

Thank you for asking this. We now show human oocytes immunostained with Ndc80 and Prc1 in the presence of colchicine (Fig. 7b). Ndc80 but not Prc1 was detected on kinetochores in colchicine-treated human oocytes.

Figure 7b

Figure 7b: Prc1 is not detectable at kinetochores. Human oocytes 5 hours post NEBD were treated with the microtubule depolymerizer colchicine and immunostained for Prc1 (green), Ndc80 (red) and DNA (Hoechst33342, blue). Scale bar, 10 μ m.

- concerning lines 258 and 341, Bennabi EMBO Rep 2018 already showed that bipolarizing too early and skipping the ball stage is critical for chromosome alignment later on. Letort Mol Biol Cell 2019 examines exactly that. These papers should be cited and discussed.

Thank you for this important suggestion. The revised manuscript now introduces the works Bennabi EMBO Rep 2018 and Letort et al Mol Biol Cell 2019, "Several microtubule regulators such as HSET, Kif11, and Tpx2, Miss, and the phosphorylation of Tacc3 are upregulated in MII

(Lefebvre et al, J Cell Biol 2002; Bennabi et al, EMBO Rep, 2018; Brunet et al, Plos One, 2018; Letort et al, Mol Biol Cell, 2019), which may contribute to rapid spindle bipolarization. Consistent with this idea, HSET overexpression accelerates spindle bipolarization in MI (Bennabi et al, EMBO Rep, 2018; Letort et al, Mol Biol Cell, 2019)” in Introduction (line 90–93). Furthermore, in Discussion, we discuss “These findings, along with previous findings that several microtubule regulators are upregulated in MII (Lefebvre et al, J Cell Biol 2002; Bennabi et al, EMBO Rep, 2018; Brunet et al, Plos One, 2018; Letort et al, Mol Biol Cell, 2019), at least partly explain why spindle formation in MI takes longer than that in MII...These notions are consistent with previous reports that accelerated spindle bipolarization, which can be induced by increased levels of HSET expression, causes chromosome alignment defects (Bennabi et al, EMBO Rep, 2018; Letort et al, Mol Biol Cell 2019)” (line 337–355).

- Line 327: the sentence should be modified, saying that these findings along others explain why spindle formation in MI takes longer than in MII. Indeed, HSET levels increase during MI (Bennabi EMBO Rep 2018), as well as Eg5 (Letort Mol Biol Cell 2019), TPX2 and P-TACC3 (Brunet Plos One 2008), MISS (Lefebvre JCB 2002) and maybe yet unknown factors...

Thank you for this suggestion. As described above, the revised manuscript introduces these relevant findings in Introduction, stating “Several microtubule regulators such as HSET, Kif11, and Tpx2, Miss, and the phosphorylation of Tacc3 are upregulated in MII (Lefebvre et al, J Cell Biol 2002; Bennabi et al, EMBO Rep, 2018; Brunet et al, Plos One, 2018; Letort et al, Mol Biol Cell, 2019), which may contribute to rapid spindle bipolarization.....The full repertoire of factors that contribute to the difference in the kinetics of spindle formation between MI and MII is unknown” (line 90–93). Moreover, in Discussion, we discuss “These findings, along with previous findings that several microtubule regulators are upregulated in MII (Lefebvre et al, J Cell Biol 2002; Bennabi et al, EMBO Rep, 2018; Brunet et al, Plos One, 2018; Letort et al, Mol Biol Cell, 2019), at least partly explain why spindle formation in MI takes longer than that in MII” (line 337–339).

Reviewer #2:

It is fast emerging that the machinery of chromosome segregation functions differently in mammalian oocyte meiosis-I. Many differences have been spotted, and are important as they may help explain the error-prone nature of this particular cell division. This paper reports that PRC1, best known as a MT crosslinker in anaphase in particular, may be found on the kinetochore in MI, as a result of a direct interaction with NDC80, and is necessary for spindle stability. This is definitely a departure from mitosis, and is thus interesting. My main hesitations are twofold. Firstly that the level of proof that the protein is kinetochore-bound needs to be higher (specific point 1 below). Secondly it is a shame that there is not a better explanation of why PRC1 is essential in MI (specific point 2 below).

We thank the reviewer for the appreciation of the interesting points of our work, and for the constructive comments.

1. The major finding that PRC1 is found on kinetochores in MI is unexpected and therefore requires further validation, particularly since it is then no longer seen on kinetochores in MII. Validation should include at least (a) immunofluorescence without a kinetochore counterstain, to rule out crosstalk/bleedthrough, and (b) demonstration using immunofluorescence that the signal disappears after PRC1 knockdown. In addition the following should be straightforward and would further support the point: (c) full length western, and (d) use of a traditional GFP tagged protein.

We thank the reviewer for asking this validation. We fully agree with the reviewer and have been very careful about the kinetochore localization of Prc1. In the revised manuscript, we now show; (a) Prc1 localization at kinetochores without a kinetochore counterstain (Fig. S6a).

Figure S6a

Figure S6a: Prc1 localizes at kinetochores in oocytes. Oocytes 2 h after NEBD were immunostained for Prc1 (anti-Prc1, green) and counterstained for DNA (Hoechst33342, blue). Scale bar, 10 μ m.

(b) A significant reduction of the kinetochore signals of Prc1 by Prc1 RNAi (Fig. S7b).

Figure S7b

Figure S7b: Reduced kinetochore Prc1 by RNAi. Oocytes microinjected with an siRNA targeting Prc1 (*siPrc1*) or luciferase (control) were fixed 2 h after NEBD. Oocytes were immunostained for Prc1 (green), ACA (kinetochores, magenta), and DNA (Hoechst33342, blue). In the plot, spots correspond to individual kinetochores and oocytes, respectively (n = 200, 200 kinetochores from 5, 5 oocytes). ****p<0.0001 by Student's t-test. Scale bar, 10 μ m. Error bars, SD.

(c) A specific band of Prc1 in a full-length Western blot (Fig. S7a).

Figure S7a

Figure S7a: Depletion of Prc1 by RNAi. Oocytes microinjected with an siRNA targeting Prc1 (siPrc1) or an siRNA targeting luciferase (control) were collected at metaphase I (6 h after NEBD) for Western blotting. Fifty oocytes were used for each sample.

(d) GFP-tagged Prc1 (Prc1-mEGFP) detected at kinetochores by immunostaining (Fig. S6e and S6f).

Figure S6

Figure S6 e and f: Prc1-mEGFP localizes at kinetochores in oocytes. Control or nocodazole-treated oocytes expressing Prc1-mEGFP were fixed at 2 h after NEBD and immunostained with anti-GFP (green) and DNA (Hoechst33342, blue). Note that we expressed Prc1-mEGFP at low levels to avoid overexpression artifacts of GFP-fused Prc1. Scale bars, 10 μm.

We found that the kinetochore signals of Prc1-mEGFP were detected only when expressed at very low expression levels followed by immunostaining. At these expression levels, kinetochore Prc1-mEGFP was below detectable levels in live oocytes. In somatic cells, EGFP-fused Prc1 is reported to be prone to overexpression artifacts that lead to excessive levels of its localization to spindle microtubules (Hu et al, MBoC, 2012, ref 66). Consistently, at higher expression levels, Prc1-mEGFP was predominantly detected along spindle microtubules (Figure for reviewer 4), which is likely an overexpression artifact. To visualize Prc1 in live oocytes, we used Prc1-SunTag at very low expression levels, which was detectable at kinetochores presumably through SunTag-mediated amplification of fluorescent signals (Fig. S6d). Detailed information of the experimental condition for visualizing fluorescently tagged Prc1 can be found in Materials and methods (line 1119–1132).

Figure for reviewer 4

Figure for reviewer 4: Prc1-mEGFP localization at high expression levels. Oocytes were microinjected with 2 pg of Prc1-mEGFP mRNA, which is a 20-fold greater amount compared to that of other experimental conditions (0.1 pg of Prc1-mEGFP or SunTag-Prc1 mRNAs). Images at early metaphase I is shown. Scale bar, 10 μ m.

Furthermore, the revised manuscript includes a new data showing that Ndc80-tethered beads recruited Prc1 when microinjected into oocytes (Fig. 4b), which further supports Ndc80-mediated recruitment of Prc1.

Figure 4b for reviewer

Figure 4b: Ndc80-bound beads associate with Prc1. Oocytes expressing mEGFP or Ndc80-mEGFP (green) were microinjected with anti-GFP beads, fixed at prometaphase I, and immunostained for Prc1 (magenta) (see Materials and methods). In the plot, spots correspond to individual beads ($n = 50$, 48 beads from 5, 5 oocytes). **** $p < 0.0001$ by Student's t-test. Scale bar, 10 μ m. Error bars, SD.

2. Assuming these simple controls validate the result – the question remains, why is it not at the kinetochore in MII? And why is it so essential in MI?

We apologize that our description has led to misunderstanding. Prc1 *does* localize at kinetochores at MII. The previous manuscript showed a z-projection image of Prc1 localization at metaphase II, in which MII-specific strong Prc1 signals along spindle microtubules overlapped with kinetochores (Fig. 2b). The revised manuscript now additionally shows a single z-slice image, in which Prc1 is clearly visible at kinetochores and strongly enriched along spindle microtubules (Fig. S6g). The revised manuscript now explicitly states “During metaphase II, Prc1 was **detected at kinetochores and strongly** enriched along microtubules in the central region of the spindle” (line 193–195). Our major conclusion, oocyte-specific kinetochore localization of Prc1 is required for MI but not for MII, remains unaffected.

Figure S6g

Figure S6g: Prc1 localizes at kinetochores and microtubules at metaphase II. One of the z-slice images of Fig. 2b at metaphase II is shown. Arrowheads indicate kinetochores. Scale bar, 10 μm .

Reviewer’s questions hold important points – (a) why Prc1 shows different localization patterns between MI and MII, and (b) why kinetochore-enriched Prc1 is essential for MI but not for MII. For both points, our data suggest that the difference in Prc1 expression levels between MI and MII is a key to answer.

(a) In MI, the low expression level of Prc1 provides a limited pool of available Prc1, which allows predominant recruitment of Prc1 to kinetochores. In MII, upregulated expression of Prc1 provides a larger pool of available Prc1, which allows its localization not only to kinetochores but also to spindle microtubules. These ideas were supported by the fact that artificially increased Prc1 expression in MI led to a significant increase in the localization levels to spindle microtubules (Fig. S12).

Figure S12

Figure S12: Increased Prc1 expression promotes its localization to spindle microtubules. Oocytes were microinjected with 2.5 or 5 pg mRNAs of Prc1. The oocytes were immunostained for Prc1 (green), ACA (kinetochores, magenta), tubulin (microtubules, gray), and DNA (Hoechst33342, blue) at metaphase I and metaphase II. Scale bar, 10 μ m.

(b) In MI, oocytes express a limited amount of Prc1, which is likely why MI requires a mechanism that locally concentrates Prc1 at kinetochores to promote spindle bipolarization. In contrast, in MII, oocytes carry upregulated levels of Prc1 and several other spindle bipolarizers (e.g. HSET, Kif11 etc), which creates a cytoplasmic environment that supports kinetochore-independent mechanisms for spindle bipolarization. These ideas are supported by the fact that MII cytoplasm rescued MI spindle bipolarization in *Ndc80*-deleted oocytes (Fig. 6a), and that artificial increase of Prc1 expression also partially rescued (Fig. 5c).

In the revised manuscript, we clearly discuss these ideas as follows.

“The limited amount of Prc1 in MI allows it to initially localize almost exclusively at kinetochores, and restricts the speed of spindle bipolarization. This idea is supported by the fact that artificially increased Prc1 localizes not only at kinetochores but also spindle microtubules (Fig. 12) and accelerates spindle bipolarization (Fig. 6)” (line 333–335).

“spindle bipolarization in MII proceeds rapidly and does not require kinetochores, as it is supported by a cytoplasmic environment that contains upregulated factors including Prc1” (line 305–307).

3. The notion that “*prc1* is one of the cytoplasmic factors that supports *NDC80*-independent spindle function” on P10 would be best supported by fusing *PRC1*-knockdown oocytes with *NDC80* knockouts.

We thank the reviewer for suggesting this experiment. To clarify, in the previous manuscript, we stated that “We hypothesized that *Prc1* is one of the cytoplasmic factors that contribute to *Ndc80*-independent spindle bipolarization in MII”. The suggested experiment would be a test for whether *Prc1* is required for supporting *Ndc80*-independent spindle bipolarization in fused oocytes.

As suggested, we performed *Prc1* RNAi in *Ndc80*-deleted oocytes, and then fused with *Ndc80*-deleted MII oocytes. We found that the RNAi procedure greatly reduced oocyte-oocyte fusion efficiency, possibly due to changes in cell surface properties during a long-term (9 hours) *in vitro* culture required for protein depletion. By repeating 9 times of experiments, we yielded only 3 available fused oocytes following control RNAi (from in total 121 oocyte pairs), and 3 available fused oocytes following *Prc1* RNAi (from in total 100 oocyte pairs). Bipolar spindle formation was observed 2/3 control fused oocytes and 0/3 *Prc1* RNAi fused oocytes (Figure for reviewer 5). Due to the scarce number of fused oocytes, we were not able to make any conclusion.

Figure for reviewer 5

Bipolar MI spindle 2 h after NEBD
Control: 2 / 3 oocytes
siPrc1 : 0 / 3 oocytes

Figure for reviewer 5: *Ndc80^{fl/fl} Zp3-Cre* oocytes at GV stage were microinjected with siRNAs and then incubated for 9 hours for protein depletion. Each of these oocytes were electrically fused to an *Ndc80^{fl/fl} Zp3-Cre* oocyte in metaphase II (MII). Spindle assembly was monitored with EGFP-Map4 (microtubules, green) and H2B-mCherry

(chromosomes, magenta). The microtubule mass that formed around MI chromosomes is magnified. Time after NEBD of the MI nucleus (h). Scale bar, 10 μ m.

Although this experiment could not give a clear answer, based on other groups' findings that MII cytoplasm contains several upregulated spindle factors (e.g. HSET, Eg5, TPX2, p-TACC3, MISS) (Lefebvre et al, J Cell Biol 2002; Bennabi et al, EMBO Rep, 2018; Brunet et al, Plos One, 2018; Letort et al, Mol Biol Cell, 2019) and that HSET overexpression accelerates spindle bipolarization in MI (Bennabi et al, EMBO Rep, 2018; Letort et al, Mol Biol Cell, 2019) as well as Prc1 overexpression, it is likely that upregulated Prc1 contributes to, but is not solely responsible for, the cytoplasmic support to kinetochore-independent spindle bipolarization in MII. This idea is consistent with our original hypothesis in the previous manuscript that "Factors that contribute to the cytoplasmic support of MII include upregulated Prc1, and possibly other upregulated microtubule regulators, such as Tpx2 and Miss".

In the revised manuscript, we clearly explain these ideas. First, we now thoroughly introduce previous important findings "Several microtubule regulators such as HSET, Kif11, and Tpx2, Miss, and the phosphorylation of Tacc3 are upregulated in MII (Lefebvre et al, J Cell Biol 2002; Bennabi et al, EMBO Rep, 2018; Brunet et al, Plos One, 2018; Letort et al, Mol Biol Cell, 2019), which may contribute to rapid spindle bipolarization. Consistent with this idea, HSET overexpression accelerates spindle bipolarization in MI (Bennabi et al, EMBO Rep, 2018; Letort et al, Mol Biol Cell, 2019)" in Introduction (line 90–93). Moreover, we have revised several statements that might have read as if MII-upregulated Prc1 is only the critical factor that supports the MII-mode of spindle bipolarization. In particular, we discuss "... MII spindles form rapidly and are independent of functional kinetochores, with support from the cytoplasm that contain upregulated spindle bipolarizers including Prc1. These findings, along with previous findings that several microtubule regulators are upregulated in MII (Lefebvre et al, J Cell Biol 2002; Bennabi et al, EMBO Rep, 2018; Brunet et al, Plos One, 2018; Letort et al, Mol Biol Cell, 2019), at least partly explain why spindle formation in MI takes longer than that in MII" (line 335–339).

4. How was aneuploidy scored, and what were the precise numbers of chromosomes? This must be stated.

Thank you for this question. We scored aneuploidy by counting the number of kinetochores in fixed MII oocytes. We imaged the entire volume of oocytes with confocal microscopy at high resolution, which allowed us to precisely count 40 kinetochores labeled with ACA (anti-centromere antibodies) in control MII oocytes.

In the revised manuscript, we now state these experimental procedures in Materials and methods (line. 1210–1213), show the entire set of z-slices images of an *Ndc80*-deleted oocytes

with the count of kinetochores (Fig. S3 and Video S4) and show plots for the precise numbers of chromosomes in *Ndc80*-deleted oocytes and Prc1-expressing oocytes (Fig. S2e and Fig. 6g).

Figure S3

Figure S3: Kinetochores counting in MII eggs. Oocytes at metaphase II were immunostained for kinetochores (ACA, green) and counterstained for DNA (Hoechst33342, magenta). Optical slice images in a z-stack of *Ndc80^{ff} Zp3-Cre* oocytes are shown. Numbers indicate kinetochores counts. Scale bar, 10 μ m.

Figure S2e

Figure S2e: Aneuploidy in *Ndc80*-deleted oocytes. Oocytes at metaphase II were immunostained for kinetochores (ACA, green) and counterstained for DNA (Hoechst33342, magenta). The number of kinetochores were counted ($n = 28$, 28 oocytes). Error bars, SD.

Figure 6g

Figure 6g: Increased *Prc1* increases aneuploidy in eggs. Oocytes at metaphase II were immunostained for kinetochores (ACA, magenta) and counterstained for DNA (Hoechst33342, blue). Arrowheads indicate a misaligned chromatid in an aneuploid egg. The number of kinetochores were counted ($n = 33$, 40 oocytes from 3 independent experiments). *** $p=0.0001$ by Student's t-test. Scale bar, 10 μ m. Error bars, SD.

5. How does *PRC1* get from the kinetochore to the microtubules – and can the authors show that happening using FRAP or some other approach?

Thank you for this question. The possibility that *Prc1* moves from the kinetochore directly to microtubules is interesting. According to the reviewer's suggestion, we considered experiments using FRAP. We reasoned that if *Prc1* molecules move from the kinetochore directly to microtubules and stably stay there, we should be able to detect cumulative fluorescence decay on spindle microtubules following photo-bleaching of kinetochore *Prc1*. However, we found that *Prc1* does not stably stay on spindle microtubules; FRAP analysis on *Prc1* along spindle microtubules showed that microtubule *Prc1* exhibits a dynamic turnover even faster than

kinetochore Prc1 (Figure for reviewer 6). Thus, FRAP approaches were unlikely to be suitable for detecting potential movement from the kinetochore to microtubules. Indeed, photo-bleaching of kinetochore Prc1 did not significantly affect Prc1 fluorescence on spindle microtubules adjacent to kinetochores (Figure for reviewer 6).

Figure for reviewer 6

Figure for reviewer 6: Oocytes expressing SunTag-Prc1 (24xGCN4-Prc1 coexpressed with scFv-sfGFP, green) at early metaphase I were used. On the top panel, SunTag-Prc1 signals were bleached at spindle microtubules (yellow circle). The recovery was monitored over time and shown in the bottom plot (black line). The data indicate that the turnover rate of Prc1 on spindle microtubules is <1 second. In contrast, after bleaching at kinetochores (circle in the lower panel), recovery at kinetochores was not observed within 5 seconds (green line in the bottom plot). Thus, turnover of Prc1 is much faster at spindle microtubules than at kinetochores. Consistently, bleaching at kinetochores (circle, lower panel) did not significantly affect Prc1 signals at kinetochore-proximal regions of spindle microtubules (squares in the lower panel; orange line in the bottom plot).

These results suggest that, although the possibility that a fraction of Prc1 moves from a kinetochore directly to a microtubule cannot be excluded, a major pool of Prc1 detected on spindle microtubules are unlikely to be directly originated from kinetochores. We thus favor the idea that a majority of Prc1 that mark spindle microtubules at metaphase I come through the cytoplasm, depending on kinetochore Prc1-dependent spindle bipolarization. Consistent with this idea, our results from a new experiment showed that Prc1 signals on kinetochore-proximal spindle microtubules are significantly decreased in monastrol-treated oocytes (Fig. 5b). These results suggest that Prc1 marks spindle microtubules at metaphase I as a consequence of spindle bipolarization.

Figure 5b for reviewer

Figure 5b: b, Kif11-dependent Prc1 enrichment along kinetochore-proximal microtubules of the bipolar spindle. Control or monastrol-treated oocytes at metaphase I (4–6 h after NEBD) were stained for Prc1 (green), microtubules (magenta) and DNA (Hoechst33342, blue). The oocytes were treated with a cold buffer for 1 minute before fixation to facilitate antibody penetration into the spindle. Prc1 signals along kinetochore-proximal microtubule bundles were measured, and their ratio to microtubule signals was calculated (n = 25, 25 locations from 5, 5 oocytes. 3 independent experiments were performed). Scale bar, 10 μ m. Error bars show the SD.

We have described these results in the revised manuscript. New statements include “**At metaphase I**, Prc1 was also enriched in the central region of the bipolar spindle, particularly along microtubule bundles, including those closely associated with kinetochores (Fig. 5b). **The Prc1 enrichment on kinetochore-proximal microtubule bundles was Kif11-dependent (Fig. 5b).** These observations suggest that Prc1 undergoes dynamic exchanges at kinetochores, and **following spindle bipolarization, it also marks kinetochore-proximal microtubule bundles**” (line 237–242). Moreover, we substantially have revised Discussion, by removing an entire paragraph including several speculative statements based on preliminary assumptions about Prc1 movement. Our major conclusion, Prc1 acts locally at kinetochores to promote spindle bipolarization, remains unaffected.

Minor points

1. *The comment ‘shift the mode of spindle bipolarisation toward that of MII’ is overstatement, and so are the experiments presented in lines 252-258*

Thank you for pointing out this. We agree that this was an overstatement unnecessary for our conclusion. We have revised the sentence. It now states “**oocytes expressing increased levels of Prc1 lack the kinetic and morphogenetic features of spindle bipolarization in MI**” (line 273–274). In the experiment following this sentence, the revised manuscript states “**Using this experimental system**, we asked whether the MI-specific mode of spindle bipolarization is critical for chromosome segregation fidelity. **When the kinetic and morphogenetic features of spindle bipolarization were altered by expression of increased levels of Prc1 in MI**, oocytes exhibited ...” (line 275–278).

2. *Some of the imaging is not quite up to this lab’s usual standards, and the surface rendering does not seem to realistically reflect the images. See for example fig 4b*

Thank you so much for finding out these problems. We found that the quality of several images was inadvertently reduced by signal interpolation effects during 3D reconstruction. The revised manuscript now shows z-projection images with no interpolation effects. Corresponding surface-rendered images are also shown (Fig. 1e, 4c, 4d, S5b, and S9c).

Figure 1e

Figure 1e: *Ndc80* Δ N/*Nuf2* Δ N rescues spindle defects. *Ndc80*^{ff} *Zp3-Cre* oocytes coexpressing *Ndc80* Δ N and *Nuf2* Δ N (*Ndc80* Δ N/*Nuf2* Δ N) and those expressing full-length *Ndc80* (*Ndc80*-WT) were monitored for spindle formation. EGFP-Map4 (microtubules, green) and H2B-mCherry (chromosomes, magenta) signals at metaphase I (5.5 h after NEBD) are shown. Scale bar, 10 μ m.

Figure 4c

Figure 4c: *Ndc80*/*Nuf2*-*Prc1* acts at kinetochores. *Ndc80*^{ff} *Zp3-Cre* oocytes expressing *Ndc80* Δ N-4A/*Nuf2* Δ N (Y564A, Q565A, L566A, and T567A mutations; indicated as 'NN Δ N-4A') or its Spc-fused form *Ndc80* Δ N-4A-Spc25C/*Nuf2* Δ N-Spc24C (indicated as 'NN Δ N-4A tethered at KTs') were monitored. Spc25C (a.a. 120–226) and Spc24C (a.a. 122–201) are kinetochores-targeting domains. Images of EGFP-Map4 (microtubules, green) and H2B-mCherry (chromosomes, magenta) at metaphase I (5.5 h after NEBD) are shown. Scale bar, 10 μ m.

Figure 4d

Figure 4d: *Ndc80ΔN/Nuf2ΔN* requires *Prc1* to promote spindle bipolarization. *Prc1* was depleted by RNAi in *Ndc80^{ff} Zp3-Cre* oocytes, and *Ndc80ΔN/Nuf2ΔN* was expressed. Images of EGFP-Map4 (microtubules, green) and H2B-mCherry (chromosomes, magenta) at metaphase I (8 h after NEBD) are shown. Scale bar, 10 μ m.

Figure S5b

Figure S5b: *Ndc80ΔN* requires *Nuf2ΔN* for spindle rescue. *Ndc80^{ff} Zp3-Cre* oocytes expressing EGFP-Map4 (microtubules, green), H2B-mCherry (chromosomes, magenta), and *Ndc80ΔN* with or without *Nuf2ΔN* were monitored. Note that spindle bipolarization was rescued only when *Ndc80ΔN* and *Nuf2ΔN* were coexpressed. Scale bar, 10 μ m.

Figure S9c

Figure S9c: Ndc80/Nuf2 requires the Prc1 binding domain for spindle bipolarization. Ndc80^{fl/fl} Zp3-Cre oocytes expressing Ndc80-Spc25C/Nuf2-Spc24C (indicated as ‘NN tethered at KT’) or Ndc80ΔC-Spc25C/Nuf2ΔC-Spc24C (indicated as ‘NNΔC tethered at KT’) were monitored. Ndc80ΔC (a.a. 1–453) and Nuf2ΔC (a.a. 1–287) lack Prc1-binding domains. Scale bar, 10 μm.

3. Some of the rhetoric and writing needs toning-down. It is an over-simplification to say that spindle assembly is kinetochore-led in oocytes, as it is known that enucleated oocytes make spindles. Similarly it is a little awkward that the authors cite the SunQY NDC80 paper but fail to mention they depleted NDC80.

Thank you for this suggestion. Having reviewers’ comments, we found that our statement “kinetochores drive spindle bipolarization” was an overstatement, since we currently lack strong evidence that the kinetochore itself is sufficient to drive spindle bipolarization. Nevertheless, our data robustly supports our conclusion that kinetochores are essential for spindle bipolarization during MI (e.g. Ndc80-deleted oocytes never establish a bipolar spindle during MI, which is rescued by targeting a Prc1-recruiting domain to kinetochores). To our knowledge, no clear evidence that enucleated oocytes make spindles at MI has been provided. Although bipolar spindle formation in enucleated oocytes has been previously reported (Brunet et al, Curr Biol 1998), this previous study enucleated metaphase I oocytes and then incubated for a long period (18 hours), so it is reasonable to consider that the bipolar spindles observed in this previous study were formed in an MII-like cytoplasmic environment. Consistent with this idea, another study (Schuh and Ellenberg, Cell 2007) reports that bipolar spindle formation was not observed within 10-12 hours following enucleation at prometaphase I.

In the revised manuscript, we have removed our statement “kinetochores drive spindle bipolarization” and similar statements that may have read as if kinetochores are sufficient to drive

spindle bipolarization. Accordingly, we have revised the title, which is now “Prc1-rich kinetochores **are required for** error-free spindle bipolarization during meiosis I in mouse oocytes”.

Also, the revised manuscript now clearly mentions that previous studies including Sun et al reported knockdown of the genes for the Ndc80 complex “In mouse oocytes, mutations and knockdowns that cause defects in kinetochore–microtubule attachment, **including knockdown of the Ndc80 complex**, can perturb spindle bipolarity (Woods et al, J Cell Biol, 1999; Sun et al, Cell Cycle 2010; Sun et al, Microscopy Microanalysis 2011; Gui et al, Dev Cell, 2013)” (line 77–79).

4. Overall the discussion is too long and several of the interpretations are over the top. For example, these results do not (alone) explain why MI is long. Overall id ask the authors to make the discussion much more concise.

Thank you for the helpful comment. As suggested, we have substantially shortened the discussion and corrected overstatements. First, we have removed an entire paragraph for discussions based on preliminary assumptions about Prc1 movement. Second, we have corrected all statements that might have read as if upregulated Prc1 is the only factor that is responsible for the cytoplasmic support for spindle bipolarization in MII. In particular, to clarify that upregulated Prc1 is not the only factor, the revised manuscript explains “**Several microtubule regulators such as HSET, Kif11, and Tpx2, Miss, and the phosphorylation of Tacc3 are upregulated in MII (Bennabi et al, EMBO Rep, 2018; Letort et al, Mol Biol Cell, 2019; Brunet et al, Plos One, 2018; Lefebvre et al, J Cell Biol 2002), which may contribute to rapid spindle bipolarization. Consistent with this idea, HSET overexpression accelerates spindle bipolarization in MI (Bennabi et al, EMBO rep, 2018; Letort et al, Mol Biol Cell, 2019)**” in Introduction (line 90–93). Furthermore, in Discussion, we discussed “**These findings, along with previous findings that several microtubule regulators are upregulated in MII (Lefebvre et al, J Cell Biol 2002; Bennabi et al, EMBO Rep, 2018; Brunet et al, Plos One, 2018; Letort et al, Mol Biol Cell, 2019), at least partly explain why spindle formation in MI takes longer than that in MII**” (line 335–339). Our major conclusion that the MII cytoplasm supports Ndc80-independent spindle bipolarization remains intact.

Reviewer #3:

This manuscript convincingly demonstrates that kinetochore recruitment of the microtubule crosslinker PRC1 by the NDC80 complex is crucial for bipolarity of the spindle in meiosis I in mouse oocytes. Furthermore, they provided interesting evidence suggesting that this kinetochore-dependent mode is important for accurate chromosome segregation, and the error-prone nature of human oocytes may be, at least partially, due to lack of PRC1 at kinetochores.

The spindle forms without centrosomes in oocytes in many animal species. How spindle bipolarity is established and maintained in oocytes is not yet understood. It is puzzling that establishment of spindle bipolarity commonly takes a long time in meiosis I in oocytes, and that spindle bipolarity is unstable in human meiosis I oocytes. The work presented in this manuscript addresses these important issues. The conclusion is novel and unanticipated, and represents a significant advance in the field of oocyte meiosis. The manuscript is well written. It is easy to follow the rationale and the flow of the work is logical.

This is an excellent piece of work, which I would be very happy to see in, say, Nature Cell Biology. It may not have a flashy message, but contains very clear conclusions and further insights into important issues related to chromosome segregation and mis-segregation in mouse and human oocytes. What I really like is how the series of experiments is constructed. Logical development of the work is excellent and multi-layers of experiments were carried out to support the conclusion. Oocytes are known to be very difficult systems to study, but these experiments presented here go well beyond what is normally expected in oocytes in terms of both quality and quantity. I found no specific concerns in the manuscript, except a minor one detailed below. I strongly recommend publication in Nature Communications.

We thank the reviewer for the appreciation of the quality and novelty of our work.

L256. Strictly speaking, these data only demonstrate that the restricted level of Prc1 is critical for slow bipolarization and error-free chromosome segregation. A causal relationship between slow bipolarization and error-free chromosome segregation is a reasonable, but unproven, hypothesis.

We thank the reviewer for this suggestion. We fully agree with the reviewer. The revised manuscript now reads “These results demonstrate that the restricted level of Prc1 **is critical for** MI-specific, slow-mode spindle bipolarization **and for** error-free chromosome segregation” (line 280–282).

REVIEWERS' COMMENTS:

Reviewer #1 (Remarks to the Author):

The points raised in the previous round of review have been satisfactorily addressed, in fact beyond my expectations. Some comments I had could have been answered without performing additional experiments, and some experiments I asked for were really challenging. However, the authors did their best to answer to ALL my comments, adding new data and convincing explanations in the text where needed. The main conclusion from the paper, the fact that the kinetochores drive spindle bipolarization in meiosis I, is now strongly reinforced. I am therefore happy to recommend publication of the revised manuscript in Nature Communications.

Reviewer #2 (Remarks to the Author):

The authors have made considerable and genuine effort to respond to the comments, and this is now a very strong piece of work.

Reviewer #3 (Remarks to the Author):

The revision has satisfactorily responded my original comment.